# Discovery of essential kinetoplastid-insect adhesion proteins and their function in *Leishmania*-sand fly interactions

Ryuji Yanase [1,7,8] ✉, Katerina Pruzinova[2], Barrack O. Owino[1], Edward Rea [1], Flávia Moreira-Leite[1,9], Atsushi Taniguchi [3,4], Shigenori Nonaka [3,5,6], Jovana Sádlová [2], Barbora Vojtkova[2], Petr Volf [2] ✉ & Jack D. Sunter [1] ✉

*Leishmania* species, members of the kinetoplastid parasites, cause leishmaniasis, a neglected tropical disease, in millions of people worldwide. *Leishmania* has a complex life cycle with multiple developmental forms, as it cycles between a sand fly vector and a mammalian host; understanding their life cycle is critical to understanding disease spread. One of the key life cycle stages is the haptomonad form, which attaches to insect tissues through its flagellum. This adhesion, conserved across kinetoplastid parasites, is implicated in having an important function within their life cycles and hence in disease transmission. Here, we discover the kinetoplastid-insect adhesion proteins (KIAPs), which localise in the attached *Leishmania* flagellum. Deletion of these KIAPs impairs cell adhesion in vitro and prevents *Leishmania* from colonising the stomodeal valve in the sand fly, without affecting cell growth. Additionally, loss of parasite adhesion in the sand fly results in reduced physiological changes to the fly, with no observable damage of the stomodeal valve and reduced midgut swelling. These results provide important insights into a comprehensive understanding of the *Leishmania* life cycle, which will be critical for developing transmission-blocking strategies.

The specific interaction and adhesion of a pathogen to a tissue is a common strategy enabling immune evasion, maintenance of infection, and life cycle progression, such as sequestration of *Plasmodium falciparum* in the vasculature[1]. By anchoring themselves, pathogens can modify their environment, with *Yersinia pestis* and the *Leishmania* parasites secreting an extracellular matrix that partially blocks the gut of their vectors leading to increased feeding and hence transmission[2–4].

The *Leishmania* haptomonad form is an important example of strong adhesion between a vector and pathogen[5]. *Leishmania* spp. are

the causative agent of leishmaniasis[6]. These flagellate parasites are transmitted between mammalian hosts by sand flies and within the fly there are six developmental stages (amastigote, procyclic, nectomonad, leptomonad, haptomonad and metacyclic promastigotes)[7–10]. Initial colonisation of the sand fly occurs through an infected blood meal after which amastigotes differentiate into motile promastigotes. These promastigotes migrate to the anterior midgut differentiating to human-infective metacyclics and haptomonads attached to the stomodeal valve[11]. Destruction of the stomodeal valve by *Leishmania*

[1]Department of Biological and Medical Sciences, Oxford Brookes University, Oxford, UK. [2]Department of Parasitology, Faculty of Science, Charles University, Prague, Czechia. [3]Laboratory for Spatiotemporal Regulations, National Institute for Basic Biology, Okazaki, Japan. [4]Research Center of Mathematics for Social Creativity, Research Institute for Electronic Science, Hokkaido University, Sapporo, Japan. [5]Spatiotemporal Regulations Group, Exploratory Research Center for Life and Living Systems, Okazaki, Japan. [6]Department of Basic Biology, School of Life Science, SOKENDAI, Okazaki, Japan. [7]Present address: School of Life Sciences, University of Nottingham, Nottingham, UK. [8]Present address: Department of Genetics and Genome Biology, University of Leicester, Leicester, UK. [9]Present address: Department of Biochemistry, Central Oxford Structural Molecular Imaging Centre (COSMIC), University of Oxford, Oxford, UK. ✉e-mail: ryuji.yanase@nottingham.ac.uk; volf@cesnet.cz; jsunter@brookes.ac.uk

results in regurgitation of parasites into the mammalian host during the next blood meal[3,12]. Understanding the molecular cell biology of these cryptic interactions is key to understanding colonisation and transmission dynamics of these important pathogens.

Haptomonad adhesion occurs via the heavily modified parasite flagellum, in which an attachment plaque is positioned adjacent to the flagellum membrane from which filaments extend towards the cell body and specific connections to the stomodeal valve are formed[13]. *Leishmania* is closely related to other kinetoplastid parasites that cause disease in humans. Adhesion of these parasites through their flagellum to their insect vector is an important part of their life cycle[4,14] and is associated with the generation of mammalian infective forms, especially for *Trypanosoma cruzi* and *T. congolense*[15,16]. Despite the importance of these attached forms the molecular makeup of them remains cryptic.

Here, we identify the first proteins of the *Leishmania* attachment plaque and show they are necessary for its assembly and the colonisation of the sand fly stomodeal valve. Moreover, loss of parasite adhesion results in fewer physiological changes to the sand fly, with reduced midgut swelling and no observable stomodeal valve damage.

Our results open up new avenues to dissect the biology of these critical forms for all kinetoplastid parasites.

## Results

### KIAPs identified through comparative proteomics

To dissect the components of the *Leishmania* haptomonad attachment plaque, we used comparative proteomics to identify proteins enriched in the attached flagellum of in vitro-derived haptomonad-like promastigotes attached to plastic (in vitro haptomonad-like promastigote)[13,17] in comparison to the flagellum of non-attached in vitro promastigotes (Fig. 1a). We identified 371 proteins with more peptide spectrum counts detected in the attached flagellum sample than the non-attached sample (Supplementary Data 1). From these 371 proteins, we only considered those proteins which were at least 8-fold enriched in the attached sample and to this resulting set of 143 proteins, we applied a series of exclusion criteria: i) proteins with an annotation or protein domain that suggested a function or localisation unrelated to adhesion e.g. metabolism, mitochondrion, transcription etc. were excluded; ii) proteins with a known function or localisation e.g. SMP1, intraflagellar transport (IFT) proteins, kinesins were

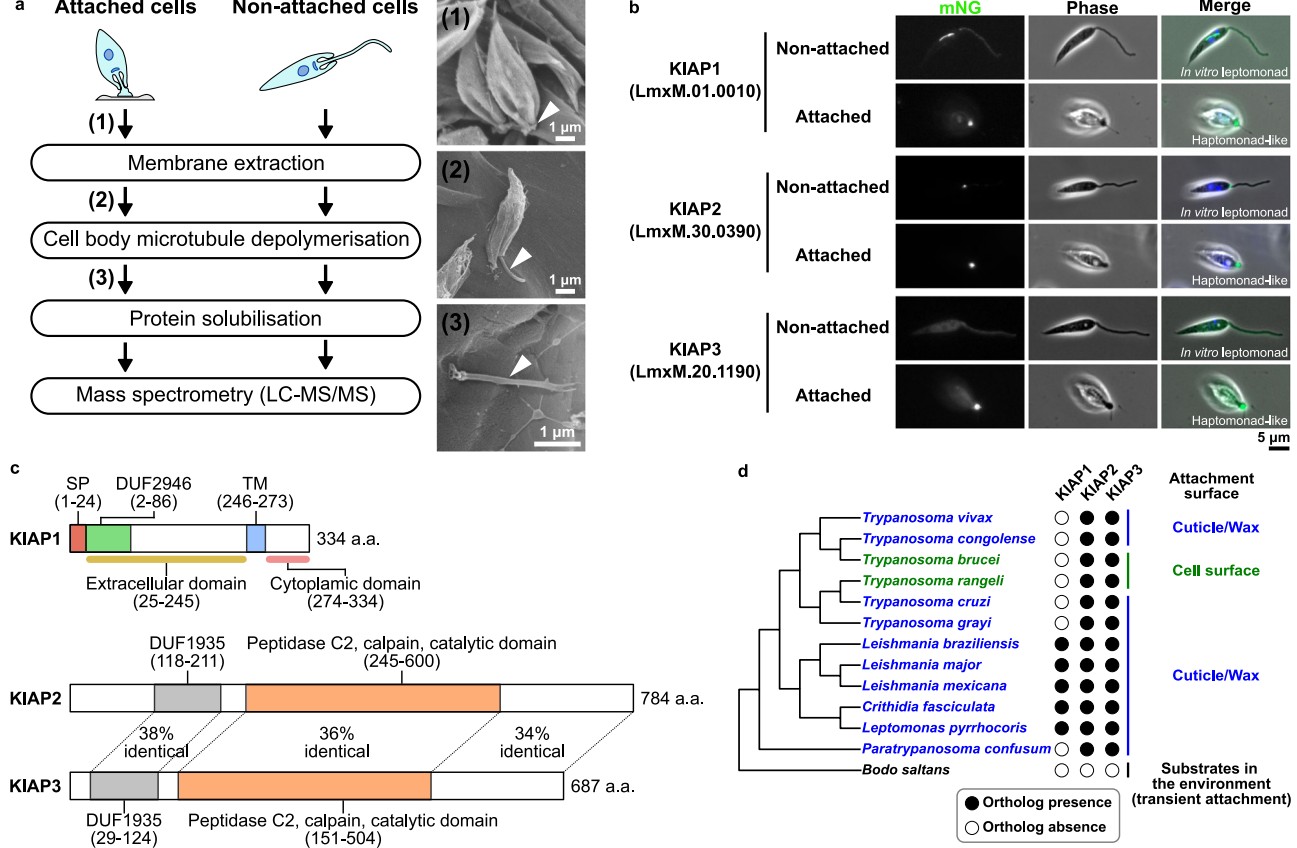

**Fig. 1 | KIAPs identified through comparative proteomics. a** Experimental flow of sample preparation for the comparative proteomic using attached in vitro haptomonad-like promastigotes and non-attached in vitro promastigotes. Scanning electron microscopy images show an intact in vitro haptomonad-like promastigote (1), an in vitro haptomonad-like promastigote after membrane extraction (2) and an attached flagellum remained on the substrate after cell body microtubule depolymerisation (3; the numbers correspond to those in the experimental flow, and each sample was prepared at the step indicated by the corresponding number). Arrowheads: attached flagellum. The scanning electron micrographs are representative of n = 2 independent sample preparations. **b** Localisation of mNeonGreen (mNG)-tagged KIAPs in non-attached in vitro promastigotes (leptomonad; see also Supplementary Fig. 2) and attached in vitro haptomonad-like promastigotes. Representative images from at least n = 3

independent sample preparations are shown. In the merged images, the overlays of the phase contrast, mNG (green) and Hoechst-stained DNA (blue) images are shown. **c** Protein domains of *L. mexicana* KIAPs. Domain names and ranges in the amino acid sequence are shown above coloured boxes. The amino acid (a.a.) length of each protein is indicated to the right of each structure. SP: signal peptide, DUF: domain of unknown function, TM: transmembrane domain. **d** Phylogenetic tree of kinetoplastids based on 18S rRNA sequences showing conservation of KIAPs across the kinetoplastids. The presence (filled circle) or absence (open circle) of an ortholog of a gene family is indicated on the right-hand side of the tree. The species indicated in blue are known to adhere to cuticle or wax layers in the insect vector and the species indicated in green are known to adhere to the cell membrane in the insect vector. *Bodo saltans* exhibits transient adhesion to substrates in the environment. Branch lengths do not represent evolutionary time.

excluded; and iii) pseudogenes and non-scaffolded genes were excluded (Supplementary Data 2). This gave a set of 39 proteins in which hypothetical proteins and those with potential cytoskeletal functions, calcium binding and cAMP signalling were prioritised[18,19]. To further refine our candidate list, we used TrypTag[20], the genome-wide protein localisation resource for the related parasite *Trypanosoma brucei* to remove proteins whose orthologs localised to organelles unlikely to be involved in adhesion such as the mitochondrion (Supplementary Data 3). 20 candidate proteins were taken forward, which we endogenously tagged with mNeonGreen (mNG) and examined their localisation in non-attached in vitro promastigotes and attached haptomonad-like promastigotes (Supplementary Fig. 1). We identified three proteins that were enriched at the enlarged tip of the attached flagellum and named these proteins Kinetoplastid-Insect Adhesion Proteins (KIAPs) 1–3 (Fig. 1b).

During the life cycle there are five different *Leishmania* promastigote stages (procyclic, nectomonad, leptomonad, metacyclic and haptomonad)[7–10]. We examined the localisation of KIAP1-3 in these different promastigote stages in a late stage in vitro culture (Supplementary Fig. 2). In the in vitro procyclic, nectomonad, leptomonad and metacyclic promastigotes KIAP1::mNG localised to the lysosome, with a weaker flagellum membrane signal and mNG::KIAP3 had a weak signal throughout the cytoplasm and the flagellum. However, in these promastigote forms, KIAP2::mNG localised to a small region within the flagellum, as it emerges from the cell body. In the in vitro haptomonad-like promastigote KIAP1::mNG, KIAP2::mNG and mNG::KIAP3 localised to the attached flagellum, with a strong signal. Given the consistency for the individual localisations of KIAP1, KIAP2 and KIAP3 in the in vitro procyclic, nectomonad, leptomonad and metacyclic promastigotes, we will refer to these different stages as non-attached in vitro promastigotes in the remainder of the manuscript.

KIAP1 (TriTrypDB gene ID: *LmxM.01.0010*) is a predicted type 1 membrane protein, with a signal peptide and transmembrane domain positioned towards the C-terminus (Fig. 1c). The N-terminus of KIAP1 contains a DUF2946 domain and the AlphaFold structural prediction of the extracellular region is similar to the C-type lectin domain of tetranectin[21] (Supplementary Fig. 3). KIAP2 and KIAP3 (TriTrypDB gene ID: *LmxM.30.0390* and *LmxM.20.1190*, respectively) have a similar domain structure, with an N-terminal DUF1935 domain and a central calpain C2 domain (Fig. 1c).

A reciprocal best BLAST analysis revealed that KIAP1 is present in all *Leishmania* species and other species including *Crithidia* and *Leptomonas* but not present in *Trypanosoma* spp., the early branching *Paratrypanosoma* and *Bodo saltans*, a free-living relative of the kinetoplastid parasite, which transiently adheres to substrates through its flagellum (Fig. 1d). KIAP2 and KIAP3 were conserved across the kinetoplastids except in *Bodo saltans* (Fig. 1d). We also confirmed that the localisation of the KIAPs in the old world species, *Leishmania major* was similar to that in *L. mexicana*, with all the KIAPs localising to the enlarged tip of the attached flagellum in the in vitro haptomonad-like promastigotes (Supplementary Fig. 4). These results suggest that the KIAPs are well conserved across the kinetoplastids, and likely represent a conserved set of components for parasite adhesion to their vectors.

## KIAPs have distinct localisation and development patterns during adhesion of in vitro haptomonad-like promastigotes

*Leishmania* adhesion proceeds through a series of defined steps, from initial adhesion, to flagellum disassembly and final maturation of the plaque[13]. To understand the point at which the different KIAPs appeared, we examined cells at different stages of adhesion (Fig. 2a–c). KIAP1::mNG in a non-attached in vitro promastigote localised to the lysosome with a weak signal along the flagellum membrane, whereas mNG::KIAP3 had a weak signal throughout the cytoplasm and the flagellum. During adhesion both proteins behaved similarly, with bright spots of KIAP1::mNG and mNG::KIAP3 observed at points along the flagellum, coinciding with membrane deformations (arrowheads; Fig. 2a, c). As the flagellum disassembled, KIAP1::mNG and mNG::KIAP3 became concentrated in the enlarged region of the flagellum adjacent to the anterior cell tip (Fig. 2a, c). In contrast, in the non-attached in vitro promastigote, KIAP2::mNG localised to a small point within the flagellum, as the flagellum emerges from the cell body (Fig. 2b). During adhesion, cells were seen in which the KIAP2::mNG signal extends along the flagellum for a short distance or an additional focus of KIAP2::mNG was observed associated with a membrane deformation (arrowhead; Fig. 2b). As the flagellum dissembled, the KIAP2::mNG signal strength increased (Fig. 2b). These results demonstrate that KIAP1, KIAP2 and KIAP3 appear at the earliest stages of parasite adhesion and likely have a role in the initial interaction with the substrate.

Next, we used time-lapse analysis to follow changes in KIAPs localisation during adhesion (Fig. 2d, e). We followed cells expressing the individual KIAPs tagged with mNG and additionally generated a cell line expressing KIAP1 endogenously tagged with mNG and KIAP3 tagged with mCherry (mCh). For KIAP1 and KIAP3, the first indication of adhesion was the localisation of these proteins to a region of flagellum membrane deformation (Fig. 2d, Supplementary Fig. 5a and Supplementary Movie 1 and 2). Over time, KIAP1 and 3 signal intensity increased at this region, with the flagellum disassembling until the mature in vitro haptomonad-like promastigote is formed. When the initial adhesion point occurred distal to the anterior cell tip both KIAP1 and KIAP3 were associated with the flagellum membrane deformation. This adhesion point remained fixed to the substrate, whilst the flagellum moved freely resulting in the movement of the cell body to this adhesion point (Fig. 2d, Supplementary Fig. 5a and Supplementary Movie 1, 2).

During adhesion, KIAP2::mNG was observed to localise along the flagellum connecting the KIAP2 focus in the flagellum at the anterior cell tip to a second KIAP2 focus that coincided with a point of membrane deformation (Fig. 2e and Supplementary Movie 3). The KIAP2::mNG focus associated with the adhesion point remained fixed to the surface and the cell body moved towards this point, with a concomitant reduction in the KIAP2 signal length between the two foci (Fig. 2e and Supplementary Movie 3). Yet, if adhesion occurred as the flagellum emerged from the cell body, KIAP2 localised only to this position (Supplementary Fig. 5b and Supplementary Movie 4). Remarkably, this reveals that during adhesion the cell body can move towards the initial adhesion point, potentially requiring a traction force.

To investigate the position of the KIAPs within the attached flagellum in more detail, we used super-resolution confocal microscopy. We generated cells expressing KIAP1 endogenously tagged with mNG and SMP1, a flagellum membrane marker, fused to mCh and examined the localisation of these proteins in attached cells (Fig. 2f). KIAP1 localised directly next to the surface on which the parasite was attached, forming a narrow band when viewed from the side and a patchy discontinuous surface when viewed from beneath. SMP1 was excluded from the attachment interface and this suggests that the membrane within the attachment plaque has a distinct composition from the rest of the flagellum membrane.

We then generated a set of cell lines with combinations of KIAPs endogenously tagged with either mNG or mCh to determine their relative positions (Fig. 2f). mCh::KIAP3 had a similar localisation to KIAP1::mNG, forming a band adjacent to the glass, with a patchy lateral distribution. While, KIAP2::mCh localised along the flagellum from adjacent to the plaque, as defined by KIAP3 localisation, to the cell body. Given the non-uniform distribution of KIAP1 and three across the attachment plaque, we revisited our electron tomograms of *Leishmania* attached in the sand fly and to plastic[13] (Fig. 2g–i and Supplementary Fig. 5c–f). We readily

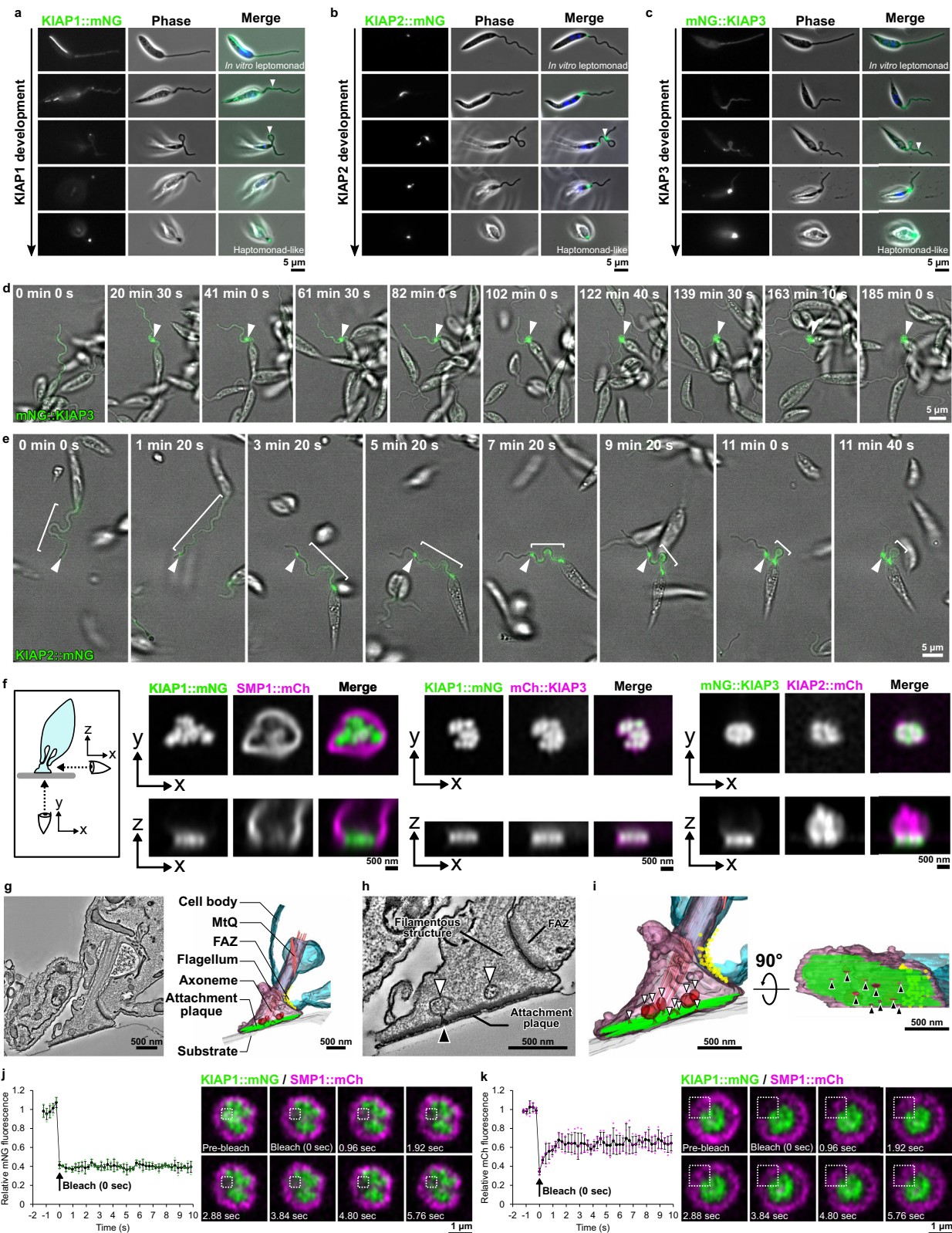

identified breaks within the attachment plaque in the tomograms, correlating with our light microscopy (Fig. 2h, i and Supplementary Fig. 5d, e). Across five plaques, 23 of the 27 breaks we found were associated with an invagination of the flagellum membrane, which likely represents endo- or exocytic vesicle fusions within the flagellum. Interestingly, the material within the vesicles in the sand fly haptomonads has a similar filamentous appearance to the

extracellular material between the parasites (Supplementary Fig. 5d, f).

Our electron tomography shows that once assembled the attachment plaque is a highly organised structure, and despite the small breaks associated with membrane dynamics, we hypothesised that there would be limited movement and turnover of the KIAPs. We used fluorescence recovery after photobleaching to investigate the

**Fig. 2 | KIAPs have distinct localisation and development patterns during adhesion of in vitro haptomonad-like promastigotes.** mNG-tagged KIAP1 (**a**), KIAP2 (**b**) and KIAP3 (**c**) development. Arrowheads denote bright spots of KIAP1::mNG or mNG::KIAP3, or additional focus of KIAP2::mNG associated with flagellum membrane deformation. Representative images from *n* = 3 independent sample preparations are shown. Merge shows phase contrast, mNG (green) and DNA (blue) signals. Sequential frames (at -20 (**d**) or -2 (**e**) min intervals) from a time-lapse movie showing in vitro haptomonad-like promastigote adhesion and mNG::KIAP3 (**d**) or KIAP2::mNG (**e**) development. Adhesion starting point (arrowheads) remain fixed. White bars show KIAP2::mNG signal range. Representative images from *n* = 2 independent sample preparations are shown. **f** Confocal microscopy of KIAP localisation in an attached flagellum includes three image sets showing KIAP1::mNG with SMP1::mCh, KIAP1::mNG with mCh::KIAP3 and mNG::KIAP3 with KIAP2::mCh (mNG: green, mCh: magenta). Representative images from *n* = 3 independent sample preparations are shown. The schematic shows viewing directions (bottom view: X–Y, side view: X–Z). **g** Serial tomogram slice and

3D reconstruction of an in vitro haptomonad-like promastigote. Representative slice from *n* = 3 different cells is shown. Vesicles near the attachment plaque (green) are shown in red. MtQ: microtubule quartet. FAZ: flagellum attachment zone. **h** Magnified slice showing vesicles (white arrowheads) near the attachment plaque and interruption of the plaque by vesicle-flagellar membrane fusion (black arrowhead). **i** Magnified 3D reconstruction showing vesicles (white arrowheads) near the attachment plaque (green) and interruption of the plaque by the vesicle-flagellar membrane fusion (black arrowheads), with side (left) and bottom (right) view. FRAP experiments of KIAP1::mNG (**j**) and SMP1::mCh (**k**), graphs show relative fluorescence intensity changes before and after photobleaching (average fluorescence intensity before photobleaching as 1). Mean ± SD (*n* = 4 independent experiments) data with green (**j**) or magenta (**k**) dots indicating values from each experiment. Photobleaching timing is marked by an arrow, and sequential frames (at -1 s intervals) of KIAP1::mNG and SMP1::mCh are displayed, with dotted white boxes indicating photobleached areas. Source data are provided as a Source Data file.

movement of KIAP1::mNG within the plaque (Fig. 2j and Supplementary Fig. 5g). After photobleaching of KIAP1::mNG, we did not observe any recovery of signal either over a short period or longer period, whereas the SMP1::mCh signal, which is restricted to the membrane outside of the plaque rapidly recovered (Fig. 2k). This shows, as predicted, that there is limited movement and turnover of KIAP1 within the attachment plaque and potentially reflects KIAP1 binding to its substrate.

### Deletion of KIAPs reduces parasite adhesion in vitro

The highly structured nature of the attachment plaque is likely the result of an ordered assembly process. To investigate the dependency relationships between the different KIAPs, we generated cell lines in which either KIAP1, KIAP2 or KIAP3 were deleted and one of the other two were endogenously tagged with mNG (Fig. 3a–c). Successful null mutant generation was confirmed by PCR (Supplementary Fig. 6a, b). The loss of any of the KIAPs led to a limited number of cells adhering, and we examined KIAP localisation in these cells. The localisation of KIAP2::mNG appeared unaffected by the loss of either KIAP1 and KIAP3, with the protein localised at the point of flagellum emergence from the cell body, with occasional extensions along the flagellum seen during adhesion (white bars; Fig. 3a, c). The deletion of either KIAP2 and KIAP3 affected the localisation of KIAP1::mNG during adhesion. Without KIAP2 and KIAP3, KIAP1::mNG did not become concentrated in a focus within the flagellum but was instead distributed throughout the flagellum and cell body, and in cells with a short flagellum there was no flagellum signal (Fig. 3b, c). In KIAP1 and KIAP2 null mutants, mNG::KIAP3 did form bright spots in the flagellum (arrowheads; Fig. 3a, b), as seen in the parental cells; however, in cells with a shorter flagellum there was little flagellum signal (Fig. 3a, b). This suggests that KIAP1 localisation is dependent on KIAP2 and KIAP3, whereas KIAP3 is initially able to localise to points of adhesion but this localisation was not stable without KIAP1 and KIAP2.

KIAP loss appeared to cause a catastrophic failure to generate mature attached cells and we sought to quantify this effect. The growth of KIAP1, KIAP2 or KIAP3 null mutants as in vitro promastigotes was unaffected (Fig. 3d), but the loss of any of the KIAPs resulted in a dramatic reduction in adhesion to glass (Fig. 3e, f). To confirm that the loss of adhesion was specific to KIAP deletion, we generated add-back cells in which a tagged version of KIAP1, KIAP2, or KIAP3 was introduced into the respective null mutant, with their expression and localisation to the enlarged flagellum tip of the in vitro haptomonad-like promastigote confirmed by fluorescence microscopy (Supplementary Fig. 6c). There was an increase in the number of attached cells observed for the add back cell lines in comparison to the null mutants (Fig. 3e, f), confirming that KIAP1, KIAP2 and KIAP3 are necessary for adhesion in vitro.

### KIAPs are essential for stomodeal valve colonisation in the sand fly

To investigate the role of KIAPs for the formation of haptomonads in the sand fly, we first infected flies with parasites expressing KIAP2::mCh and mNG::KIAP3 or KIAP1::mNG and SMP1::mCh, and examined the expression of these proteins in dissected midguts on day 8 post-blood meal (PBM; Fig. 4a and Supplementary Fig. 7a). KIAPs were clearly expressed on parasites that had colonised the surface of the stomodeal valve, with a strong fluorescence signal outlining the valve. Moreover, there was little signal from the other parasites located in the cardia region.

Next, we infected flies with the parental, null mutant and added back cell lines for KIAP1, KIAP2 and KIAP3, and examined the distribution of parasites on day 6 and 9 PBM (Fig. 4b–d). There were minimal differences in the infection rates and loads between the different cell lines and all developed heavy late-stage infections (Fig. 4b and Supplementary Fig. 7b); however, large differences were found in the location of the parasites (Fig. 4c). While 70–90% of sand flies had the stomodeal valve colonised by the parental and add back cells by day 9 PBM, no colonisation of the valve was observed for the KIAP1, KIAP2 and KIAP3 null mutants. The null mutants migrated to the cardia region but did not attach to the stomodeal valve.

We imaged the valves from infected sand flies (Fig. 4d). The parasites expressed the flagellum membrane marker SMP1 fused to mCherry, enabling us to determine their position. For both the parental and add back cells *Leishmania* parasites were associated with the stomodeal valve, whereas for the KIAP1, KIAP2 and KIAP3 null mutants there was a near complete loss of SMP1::mCh signal associated with the valve (Fig. 4d). Overall, this shows that KIAP1, KIAP2 and KIAP3 are necessary for colonisation of the stomodeal valve in the sand fly, correlating with the loss of adhesion in vitro.

### Infection with KIAP null mutants causes reduced physiological damage to the sand fly

A late stage infection of sand flies is characterised by the generation of the mammalian infective form (metacyclic), damage to the stomodeal valve cuticle surface and the generation of the promastigote secretory gel that blocks the midgut causing it to swell[3,22,23]. We first analysed the percentage of metacyclics based on their morphology[24] present on day 9 PBM for the parental, KIAP1, KIAP2 and KIAP3 null mutants and respective add back parasites (Supplementary Fig. 7b, c). Overall, there was limited difference in the mean percentage of metacyclic forms observed between the mutants and the parental parasites. Moreover, we were readily able to find nectomonad, leptomonad, metacyclic forms for all the cell lines examined and haptomonads in the parental and add back parasites (Supplementary Fig. 7d).

To assess the pathology the *Leishmania* parasites caused in the sand fly, we examined the structure of the thoracic midgut and cardia

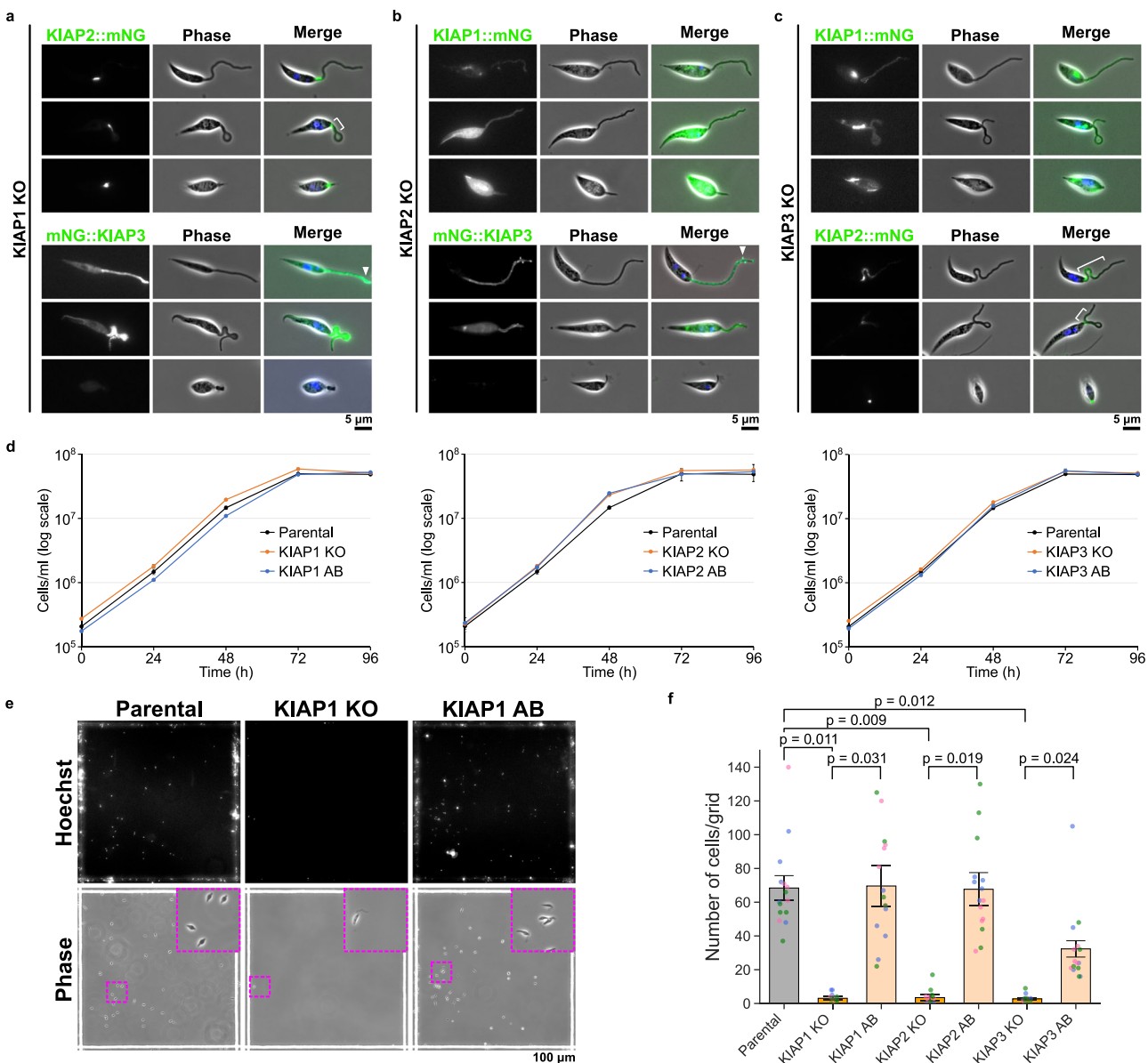

**Fig. 3 | Deletion of KIAPs reduces parasite adhesion in vitro.** Localisation of mNG-tagged KIAPs in KIAP1 (**a**), KIAP2 (**b**) or KIAP3 (**c**) KO *L. mexicana* cell lines. Representative images from *n* = 2 independent sample preparations are shown. In the merged images, the overlays of the phase contrast, mNG (green) and Hoechst-stained DNA (blue) images are shown. White bars: extensions of mNG::KIAP2 along the flagellum. Arrowheads: bright spots of mNG::KIAP3. **d** Growth curves of parental and KIAP1 (left), KIAP2 (middle) and KIAP3 (right) gene knockout (KO) and add back (AB) cell lines. Data represent mean ± SD (*n* = 3 independent experiments). Source data are provided as a Source Data file. **e** Hoechst fluorescence and phase images of parental, KIAP1 KO and AB *Leishmania* cells induced adhesion for 24 h on a gridded glass coverslip. Representative images from *n* = 3 independent sample preparations are shown. Top; Hoechst images. Bottom; phase images. Insets show a magnified view of the magenta dotted square area. **f** Quantification of the number of attached cells per grid area for parental and KIAPs KO and AB *L. mexicana* cell lines. Data represent mean ± SEM (*n* = 3 independent experiments). The blue, pink and green dots represent measurement from three independent experiments, respectively. *P*-values were calculated using two-tailed Welch's *t*-test, which are shown above the graph. Source data are provided as a Source Data file.

region of the midguts from sand flies infected with parental parasites, KIAP1, KIAP2 and KIAP3 null mutants and the corresponding add backs by light microscopy. We noted that the midguts infected with the parental parasites and the add backs were more swollen and wider than those infected with the null mutants even though the number of parasites in the midgut was almost same between the mutants and the parental parasites (Fig. 5a, b and Supplementary Fig. 7b). To investigate the infected midguts further, we captured movies and examined the movement of the parasites with and without disruption of the midgut (Fig. 5a and Supplementary Movie 5–11). In all movies, the thoracic midgut was heavily infected with parasites. The parental and add-back parasites were not able to move freely and when the midgut was

ruptured the parental and add back parasites were embedded in the promastigote secretory gel. However, in the midguts infected with the KIAP1, KIAP2 and KIAP3 null mutants, the parasites were much more motile and when the midgut was ruptured the parasites swam freely and little promastigote secretory gel was seen.

Finally, we stained the cuticle surface of the stomodeal valve with calcfluor, a chitin stain (Fig. 5c). In non-infected sand fly midguts, calcfluor distinctly stained the chitin layer of the stomodeal valve and also the crop. Interestingly, a similar pattern of calcfluor staining was observed in midguts infected with the KIAP1-3 null mutants, with a distinctive signal from the chitin layer. In contrast, in midguts from sand flies infected with the parental parasites or the add-backs, the

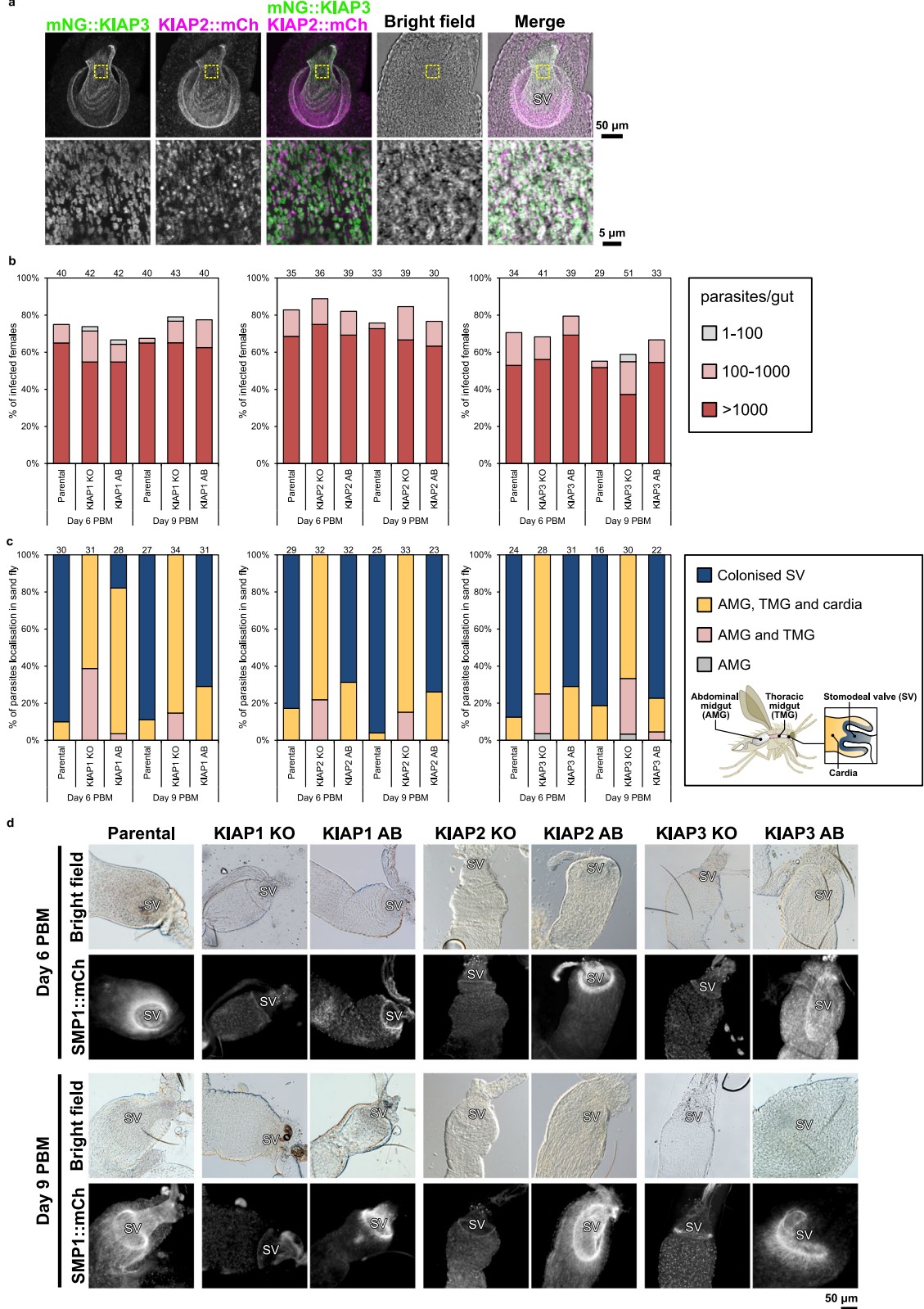

stomodeal valve was distorted with a much reduced calcofluor signal (Fig. 5c). Overall, this suggests that infection of sand flies with the KIAP null mutants results in less damage to the midgut.

## Discussion

*Leishmania* has multiple developmental forms in the sand fly, including the cryptic haptomonad form that is attached to the cuticle surface of the stomodeal valve in the anterior midgut. Haptomonad adhesion is mediated through a modified and shortened flagellum that contains a highly organised cytoskeletal structure that extends through the flagellum from the cell body to the attachment plaque[5,13]. Here, we have identified the first proteins that are essential for haptomonad adhesion to the sand fly stomodeal valve.

**Fig. 4 | KIAPs are essential for stomodeal valve colonisation in the sand fly.**
**a** Z-stack confocal microscopy analysis of KIAP localisation in a dissected sand fly gut on day 7 post blood meal (PBM) using *L. mexicana* cells expressing mNG::KIAP3 and KIAP2::mCh. The confocal fluorescence images are shown as a maximum intensity projection of a series of confocal z-sections (8 × 2.13 μm z-step). The location of the stomodeal valve is indicated by SV. Representative images from *n* = 4 infected midguts. **b** Infection rates and intensities of infections of parental and KIAP1 (left), KIAP2 (middle) or KIAP3 (right) KO and AB *L. mexicana* cells on day 6 and 9 PBM. Numbers above the bars indicate the number of dissected female sand flies. Infection intensities were evaluated as light (<100 parasites/gut; grey), moderate (100–1000 parasites/gut; pink) and heavy (>1000 parasites/gut; red). The cumulative percentages obtained from two independent experiments are shown in

the graphs. Source data are provided as a Source Data file. **c** Localisation of infections of parental and KIAP1 (left), KIAP2 (middle) or KIAP3 (right) KO and AB *L. mexicana* cells on day 6 and 9 PBM. Numbers above the bars indicate the number of evaluated (positive) female sand flies. The schematic diagram shows the name and location of each part of a sand fly gut. The cumulative percentages obtained from *n* = 2 independent experiments are shown in the graphs. Source data are provided as a Source Data file. **d** Bright field and fluorescence images of dissected sand fly guts on day 6 (left) or 9 (right) PBM where parental, or KIAP1, KIAP2 or KIAP3 KO and AB *L. mexicana* cells expressing SMP1::mCh were infected. Representative images of infected midguts of the *n* = 2 independent infections from (**b, c**). The sand fly cartoon in Fig. 4c was kindly provided by Dr Richard Wheeler (University of Oxford).

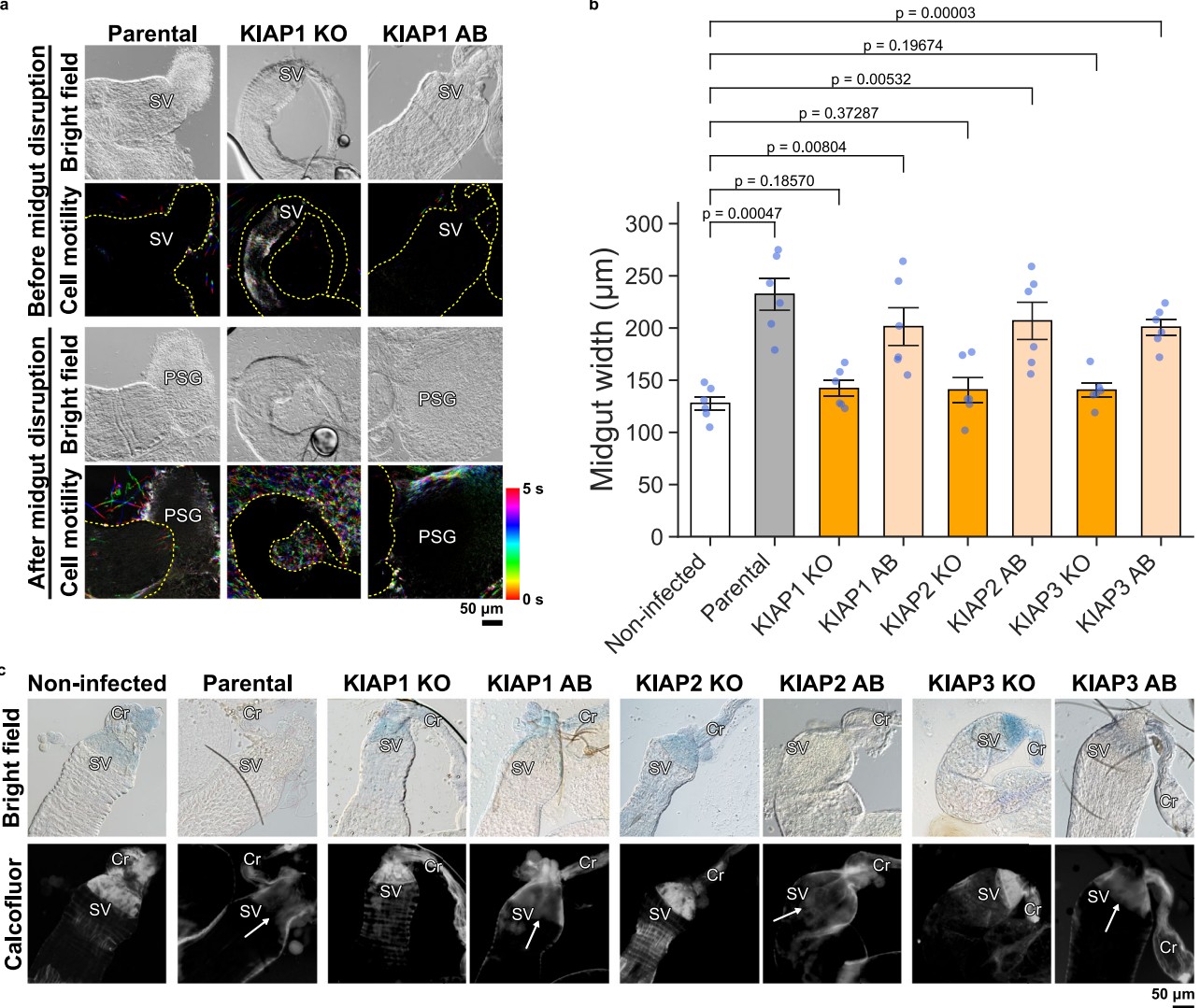

**Fig. 5 | Infection with KIAP null mutants causes reduced physiological damage to the sand fly. a** Bright field images extracted from movies of dissected sand fly guts infected with parental, or KIAP1 KO and AB *L. mexicana* parasites before and after midgut disruption on day 8 PBM. The temporal-colour-coded cell tracks were generated from the movies (Supplementary Movies 5–7; see Methods) capturing five seconds of the *L. mexicana* cell motility within the midgut or around the midgut and PSG after midgut disruption. Regions highlighted in colour or white indicate tracks of *Leishmania* motility. Note that the midguts infected with the parental and the KIAP1 AB cells are more swollen and wider than that infected with the KIAP1 KO cells. The locations of the stomodeal valve and promastigote secretary gel are shown with SV and PSG, respectively. The yellow dotted line shows the outline of the midgut. Representative movies of *n* = 2 independent infections.

**b** Quantification of width of dissected midguts non-infected or infected with parental, or KIAP1, KIAP2 or KIAP3 KO and AB *L. mexicana* parasites on day 9 PBM. The width of the widest part under the stomodeal valve was measured. Data represent mean ± SEM (*n* = 6 individually infected midguts). The blue dots represent each measurement. P-values were calculated using two-tailed Welch's *t*-test, which are shown above the graph. Source data are provided as a Source Data file. **c** Bright field and fluorescence images of dissected and calcofluor-stained midguts non-infected or infected with parental, or KIAP1, KIAP2 or KIAP3 KO and AB *L. mexicana* parasites on day 7 PBM. The arrow points to the region of low calcofluor signal. The locations of the stomodeal valve and crop are shown with SV and Cr, respectively. Representative images of at least *n* = 10 infected midguts.

We took advantage of a simple yet powerful comparative proteomic approach coupled with an endogenous tagging screen to identify KIAP1-3. This screening approach has caveats and will not provide a complete component list for the attachment plaque, with more proteins likely still to be discovered. For example, the use of detergent will likely remove membrane associated proteins, though clearly not all, as KIAP1 is a predicted a membrane protein. Moreover, we generated our non-attached flagellum sample from cells that had not been given the opportunity to adhere rather than the non-attached cells that are seen after the in vitro adhesion procedure. The latter cells are likely to contain adhesion intermediates but had yet to stably attach. A triangulation of these three flagellar samples will likely identify additional candidates.

KIAP1 is a transmembrane domain protein, with an extracellular domain that has a predicted 3D structure similar to that of a C-type lectin, whereas KIAP2 and KIAP3 have a central calpain-like domain. The calpain domain is found in many other important cytoskeletal proteins in the trypanosomatids and is catalytically inactive as it lacks the critical cysteine residue[25–27]. The calpain-like domain is therefore likely important for mediating protein-to-protein interactions. The presence of the C-type lectin-like extracellular domain in KIAP1 suggests that a carbohydrate moiety plays a role in mediating *Leishmania* adhesion. The origin of this carbohydrate is unclear; within the sand fly, the cuticle contains N-acetyl-glucosamine but this is potentially inaccessible below its wax surface, while in culture the serum may provide carbohydrate for adhesion. *Leishmania* also secretes proteophosphoglycans[28] and these could act as the substrate for KIAP1 to bind. A similar strategy is used by fungal plant and insect pathogens, where they secrete substrates which they use to anchor themselves to the cuticle[29]. However, in vitro *Leishmania* adheres to hydrophobic substrates and may form adhesion through hydrophobic interactions on the waxy surface of the stomodeal valve, as suggested for *T. cruzi* adhesion[30].

The KIAPs are well conserved across the kinetoplastids, and likely represent a conserved set of components for parasite adhesion to their vectors. Both, KIAP2 and KIAP3 are conserved across the species we examined but are not present in *Bodo saltans*, which is free-living. *B. saltans* transiently adheres to surfaces through its flagellum but this adhesion is not associated with the assembly of an attachment plaque[31,32]; thus, these transient adhesive events are unlikely to require KIAP2 and KIAP3. KIAP1 had a more restricted evolutionary profile and was only observed in *Leishmania* and closely related species. This is unsurprising as KIAP1 is likely important for interacting with molecules on the surface of the insect tissues and will therefore have evolved in concert with its ligand. It is likely that the other kinetoplastid species have a protein that performs the equivalent function of KIAP1 either a version that is too divergent to identify by BLAST or its function is performed by a different protein.

We have previously shown that the modified flagellum of the haptomonad contains multiple different structural elements, including filaments that extend from the cell body to the attachment plaque, with connections across the membrane that interact with the stomodeal valve surface[13]. Our co-localisation analysis, suggested that both KIAP1 and KIAP3 are associated with the attachment plaque and given that KIAP1 is a transmembrane protein, it likely forms part of the connections to the stomodeal valve. KIAP2 had a distinct localisation, forming an elongated structure that extended from the attachment plaque towards the cell body and is therefore potentially a component of the filaments within the modified flagellum.

Given the complexity of the attachment structure, its assembly likely proceeds in an ordered manner, with integration of different components at different times. KIAP1-3 were all present in the flagellum at the earliest stages of adhesion, suggesting these are core components, which is reflected in their essentiality for parasite adhesion. All the KIAP null mutants attached at a very low rate in vitro, which enabled us to examine their dependency relationships. The localisation of the membrane protein KIAP1 was dependent on the expression of KIAP2 and KIAP3, whereas KIAP3 is initially able to localise to points of adhesion, but this localisation was not stable without KIAP1 or KIAP2. KIAP2 remain associated with the flagellum next to the cell body when either KIAP1 or KIAP3 were deleted. This suggests that KIAP2 localisation is not dependent on KIAP1 and KIAP3 expression; however, as KIAP2 is found in this position in the in vitro promastigote this is perhaps not surprising. It should be noted though that as the KIAP null mutants are generally unable to attach, the organisation of the attachment plaque is potentially disrupted. Our data is therefore informative of whether the KIAPs are able to localise to regions within a flagellum that is attempting to attach; however, this is unlikely to fully reflect the situation within an attaching parental cell.

There are two distinct types of *Leishmania* adhesion within the sand fly. Firstly, the flagellum of the nectomonad interdigitates between the midgut microvilli and secondly, the highly modified flagellum of the haptomonad attaches to the stomodeal valve or pylorus region of the hindgut[10]. Haptomonad adhesion to the hindgut is restricted to *Leishmania* species in the *Viannia* subgenus such as *L. braziliensis*[10]; whereas haptomonad adhesion to the stomodeal valve is ubiquitous across human-infective *Leishmania*. Nectomonad adhesion to the midgut microvilli ensures the parasite is not excreted with the blood meal remnants[33], and as the KIAP null mutants were able to migrate to the anterior midgut this adhesion still occurs. The KIAP null mutants were tested in a permissive sand fly species (*L. longipalpis*) in which nectomonad adhesion relies on the interaction between the parasite surface and midgut mucin[33]. There are specific sand fly and *Leishmania* species pairings (e.g. *L. major* and *P. papatasi*[34]) in which nectomonad adhesion relies on the LPG terminal galactose and the sand fly galectin. The functional predictions for KIAP1-3 do not suggest a role in LPG synthesis, so it is likely that even in restrictive sand flies the KIAPs are not necessary for nectomonad adhesion.

Furthermore, the dynamics and ultrastructural changes associated with the two types of *Leishmania* adhesion are distinct, suggesting they occur through distinct mechanisms. The nectomonad flagellum and microvilli interaction is not associated with any ultrastructural changes to the flagellum and is reversible, enabling the onward migration of the parasite; whereas, haptomonad adhesion requires flagellum remodelling forming a strong and stable connection. However, despite these distinctions the initial interaction between the *Leishmania* parasite and the stomodeal valve will be mediated by the parasite surface and may have commonalities with nectomonad adhesion. In addition, the capacity of haptomonads to differentiate into leptomonads/metacyclics is unknown but the stability of the attachment plaque suggests that the attached cell is terminally differentiated. Interestingly, we previously observed that ~1% of haptomonads were dividing[13], which may generate a free-swimming daughter and such asymmetric differentiation divisions are found in other kinetoplastids[35].

In the sand fly infections, none of the KIAP null mutants colonised the stomodeal valve, yet they had migrated to the anterior midgut and were able to differentiate into metacyclics. This indicates that the loss of adhesion is a specific defect rather than an issue with parasite viability and infectivity within the sand fly. Given this specific effect, we investigated key features of a late-stage sand fly infection including distention of the midgut by the promastigote secretory gel and stomodeal valve cuticle damage. Previous work has shown that the combination of the promastigote secretory gel blocking the midgut and damage to the stomodeal valve through the action of chitinase enhances parasite transmission through increased sand fly feeding attempts[12,23], while the promastigote secretory gel has been shown to enhance establishment of *Leishmania* infection[23]. Remarkably, the KIAP null mutants were motile within the midgut in contrast to the parental cells, likely due to the limited amount of promastigote

secretory gel in these midguts. This aligned with the reduction in midgut width of those sand flies infected with the KIAP null mutants. Moreover, the chitin staining of the stomodeal valve in the sand flies infected with the KIAP null mutants closely resembled that of the uninfected sand fly, with no observable damage. Based on these observations, we anticipate there will be a reduced transmission rate to and/or pathology in the mammalian host with the KIAP null mutants. Moreover, preliminary work from Catta-Preta et al. has shown that haptomonads alone are capable of causing lesions in mice and when co-inoculated with a low dose of late metacyclics, they exacerbated pathology[9]. However, we acknowledge that *Leishmania* transmission is a multifactorial process and incompletely understood, so making such predictions is not straightforward. The KIAP null mutants were still competent to differentiate into metacyclics and have therefore the potential to infect a mammalian host.

In conclusion, we have identified and functionally characterised the first proteins necessary for the generation of the attachment plaque and adhesion of *Leishmania* to the stomodeal valve in the sand fly. The KIAPs were predominantly expressed in the haptomonads and are therefore the first molecular markers of this form, enabling and informing future analyses. The conservation of these proteins across the kinetoplastids suggests a common mechanism of adhesion and attachment plaque generation, and provides an important insight into our understanding of the development and differentiation of these parasites in their insect vectors, which will be critical for the development of transmission-blocking strategies.

## Methods

### Cell culture

Cas9T7 *L. mexicana* (WHO strain MNYC/BZ/1962/M379, expressing Cas9 and T7 RNA polymerase) and *L. major* Friedlin (pTB007 cell line, expressing Cas9 and T7 RNA polymerase[36]) promastigotes were grown at 28 °C in M199 medium (Gibco, UK) with 10% foetal calf serum, 40 mM HEPES-HCl (pH 7.4), 26 mM NaHCO$_3$ and 5 μg/ml haemin. Cells were maintained in logarithmic growth by regular subculturing.

### In vitro haptomonad-like promastigote adhesion

Axenic in vitro haptomonad-like promastigotes were generated by culturing $1 \times 10^6$ cells/ml promastigotes on 13 mm round Thermanox plastic coverslips (Nalgene Nunc International, Rochester, NY) scratched with sandpaper and sterilised with 100% ethanol (for proteomics and electron microscopy) or gridded glass coverslips grid-500 (iBidi, Gräfelfing, Germany) which were cut into small pieces of ~5 × 5 mm and sterilised with 100% ethanol (for widefield epifluorescence microscopy) in a 24 well plate with 1 ml of M199 medium at 28 °C with 5% CO$_2$ for 72 h with M199 medium being replaced every 24 h (for proteomics and electron microscopy) or for 24 h (for widefield epifluorescence microscopy).

### Comparative proteomics

For preparation of in vitro haptomonad-like promastigote attached flagellum sample, twenty-four Thermanox plastic coverslips on which *Leishmania* had attached were prepared in a 24 well plate. They were washed with 1 ml of Voorheis's modified PBS (vPBS; 137 mM NaCl, 3 mM KCl, 16 mM Na$_2$HPO$_4$, 3 mM KH$_2$PO$_4$, 10 mM glucose, 46 mM sucrose, pH 7.6) three times and incubated in 1% (w/v) IGEPAL (Sigma-Aldrich, UK) in PEME (0.1 M PIPES (pH 6.9), 2 mM EGTA, 1 mM MgSO$_4$, 0.1 mM EDTA) with cOmplete, Mini Protease Inhibitor Cocktail without EDTA (Roche, Basel, Switzerland) for 5 min. Then coverslips were washed with 1 ml of a fresh 1% (w/v) IGEPAL in PEME with protease inhibitor cocktail and incubated in 1 ml of 300 mM CaCl$_2$ in PEME with protease inhibitor cocktail for 2 min. Coverslips were washed with PEME twice. For 24 coverslips, proteins were solubilised with 100 μl of a solubilisation buffer (2% (w/v) SDS, 60 mM Tris-HCl (pH 6.8), 50 mM

DTT) with protease inhibitor cocktail per 6 coverslips and a total of ~500 μl of attached flagellum sample was collected.

For promastigote non-attached flagellum sample preparation, $8 \times 10^7$ promastigote cells were harvested by centrifugation (800 g for 7 min) and washed twice with 5 ml and 1 ml of vPBS. Cells were then incubated in 150 μl of 1% (w/v) IGEPAL in PEME with protease inhibitor cocktail and centrifuged (17,000 g for 2 min). The precipitate was resuspended in 150 μl of 300 mM CaCl$_2$ in PEME with protease inhibitor cocktail and centrifuged (17,000 g for 10 min at 4 °C). The precipitate was solubilised in 400 μl of a solubilisation buffer with protease inhibitor cocktail.

For mass spectrometry analysis, the flagellum samples of the attached in vitro haptomonad-like promastigote and non-attached in vitro promastigote were digested with sequencing grade trypsin (Promega, Southampton, UK) following, alkylation with iodoacetamide, immobilisation and clean up on suspension trap (S-Trap; ProTifi, Fairport, NY), following the manufacturer's recommended protocol. Digestion proceeded overnight at 37 °C before peptides were eluted, dried down and resuspended in aqueous 0.1% TFA for LC-MS/MS.

Peptides were loaded onto an mClass nanoflow UPLC system (Waters, Wilmslow, UK) equipped with a nanoEaze M/Z Symmetry 100 Å C$_{18}$, 5 μm trap column (180 μm × 20 mm, Waters) and a PepMap, 2 μm, 100 Å, C$_{18}$ EasyNano nanocapillary column (75 μm x 500 mm, Thermo Fisher Scientific, Waltham, MA). The trap wash solvent was aqueous 0.05% (v/v) trifluoroacetic acid and the trapping flow rate was 15 μL/min. The trap was washed for 5 min before switching flow to the capillary column. Separation used gradient elution of two solvents: solvent A, aqueous 0.1% (v/v) formic acid; solvent B, acetonitrile containing 0.1% (v/v) formic acid. The flow rate for the capillary column was 300 nL/min and the column temperature was 40 °C. The linear multi-step gradient profile was: 3–10% B over 7 mins, 10–35% B over 30 min, 35–99% B over 5 min and then proceeded to wash with 99% solvent B for 4 min. The column was returned to initial conditions and re-equilibrated for 15 min before subsequent injections.

The nanoLC system was interfaced with an Orbitrap Fusion Tribrid mass spectrometer (Thermo Fisher Scientific) with an EasyNano ionisation source (Thermo Fisher Scientific). Positive ESI-MS and MS$^2$ spectra were acquired using Xcalibur software (version 4.0, Thermo Fisher Scientific). Instrument source settings were: ion spray voltage, 1900 V; sweep gas, 0 Arb; ion transfer tube temperature; 275 °C. MS$^1$ spectra were acquired in the Orbitrap with: 120,000 resolution, scan range: *m/z* 375–1500; AGC target, 4e$^5$; max fill time, 100 ms. Data-dependant acquisition was performed in top speed mode using a 1 s cycle, selecting the most intense precursors with charge states >1. Easy-IC was used for internal calibration. Dynamic exclusion was performed for 50 s post precursor selection and a minimum threshold for fragmentation was set at 5e$^3$. MS$^2$ spectra were acquired in the linear ion trap with: scan rate, turbo; quadrupole isolation, 1.6 *m/z*; activation type, HCD; activation energy: 32%; AGC target, 5e$^3$; first mass, 110 *m/z*; max fill time, 100 ms. Acquisitions were arranged by Xcalibur to injections for all available parallelizable time.

Tandem mass spectra were extracted using MSConvert to.mgf format before submitting to database searching using Mascot (Matrix Science, version 2.7.0.1). Mascot was set up to search the *Leishmania mexicana* subset of the TriTrypDB database (https://tritrypdb.org/tritrypdb/app)[37] appended with common proteomic contaminants (number of proteins in database = 8368). Data were searched with a fragment ion mass tolerance of 0.50 Da and a parent ion tolerance of 3.0 ppm. O-124 of pyrrolysine, j-16 of leucine/isoleucine indecision and carbamidomethyl of cysteine were specified in Mascot as fixed modifications. Oxidation of methionine was specified in Mascot as a variable modification. Scaffold (version Scaffold_5.2.0, Proteome Software Inc.) was used to validate MS/MS-based peptide and protein identifications. Peptide identifications were accepted if they could be established at greater than 10.0% probability to achieve an FDR less than

1.0% by the Percolator posterior error probability calculation[38]. Protein identifications were accepted if they could be established at greater than 98.0% probability to achieve an FDR less than 1.0% and contained at least 2 identified peptides. Protein probabilities were assigned by the Protein Prophet algorithm[39]. Proteins that contained similar peptides and could not be differentiated based on MS/MS analysis alone were grouped to satisfy the principles of parsimony. Proteins sharing significant peptide evidence were grouped into clusters.

Our proteomic analysis was based on a single biological replicate, with an additional localisation screen to ensure only haptomonad-associated proteins were further analysed. To determine the relative change in protein abundance between the samples, we compared the total spectrum counts for each protein, which is a measure of protein abundance within the sample. In our dataset, we examined the change in abundance of 10 axonemal proteins (LmxM.13.0430, LmxM.27.0520, LmxM.08_29.0660, LmxM.20.1400, LmxM.36.6380, LmxM.31.0230, LmxM.10.1190, LmxM.23.1310, LmxM.33.3880, LmxM.24.1030), as we expected minimal changes in abundance for these proteins (Supplementary Data 1). The change in abundance of these proteins varied from 0.5 to 2-fold between the samples, confirming our expectations and the quality of the dataset. Among the proteins identified in the flagellum samples of the attached in vitro haptomonad-like promastigote and non-attached in vitro promastigote samples, only those that were specific (129/143) or 8-fold more abundant (14/143) in the in vitro haptomonad-like promastigote sample were considered (Supplementary Data 1 and 2). A set of three exclusion criteria were applied: i) proteins with an annotation or protein domain that suggested a function unrelated to attachment e.g. metabolism, mitochondrial, protein folding and assembly, endocytic trafficking, transcription, transporters, proteasome, cell cycle kinase were excluded. ii) proteins with a known function or localisation e.g. SMP1, IFT proteins, kinesins were excluded. iii) pseudogenes and non-scaffolded genes were excluded (Supplementary Data 2). The resulting 39 proteins were then further refined by analysing information about the *T. brucei* orthologs–23 of the 39 proteins had orthologs. The 16 proteins without orthologs were taken forward. If the *T. brucei* ortholog of the *Leishmania* protein had known functions or localisations that would unlikely be involved in haptomonad adhesion were excluded e.g. mitochondrial (Supplementary Data 3). The 20 proteins in the list thus selected were then endogenously tagged with mNeonGreen and their localisation in a non-attached in vitro promastigote and an attached in vitro haptomonad-like promastigote was examined.

## Scanning electron microscopy

Intact in vitro haptomonad-like promastigote, or haptomonad-like promastigote after being treated with 1% IGEPAL in PEME or with 1% IGEPAL and 300 mM $CaCl_2$ in PEME on Thermanox plastic coverslips were fixed respectively with 2.5% glutaraldehyde in PEME. After an hour, coverslips were washed once in PEME and once in $ddH_2O$. The coverslips were then dehydrated using increasing concentrations of ethanol (30%, 50%, 70%, 90%, 100% (v/v) and 2 × absolute ethanol; 10 min/step)). The coverslips were then critical point dried, mounted onto SEM stubs using carbon stickers, and sputter coated with a layer of 12–14 nm of gold. Images were taken on a Hitachi S-3400N scanning electron microscope at 5 kV, at a 5.5 mm working distance.

## Transfection and drug selection

Generation of *L. mexicana* and *L. major* tagging constructs and sgRNA templates for endogenous mNeonGreen or mCherry tagging were generated by the PCR method as previously described using pLPOT (mNG/Puromycin) or pLPOT (mCh/Neomycin) as the template, respectively[40]. Transfection of cells was performed as previously described using the Amaxa Nucleofector-2b[41] Primers for constructs

and sgRNA were designed using LeishGEdit (http://www.leishGEdit.net)[42]. For the generation of KIAP1, 2 and 3 null mutants using CRISPR/Cas9 mediated genome editing, the C9/T7 cell line was transfected with guide and repair constructs generated by PCR using primers designed on the LeishGEdit using the G00 primer and the pTBlast and pTPuro plasmids as templates[42]. Constructs were transfected using a Nucleofector 2b. Successful transfectants were selected with 20 μg/ml Puromycin (Melford Laboratories, Ipswich, UK) or 20 μg/ml G-418 (Melford Laboratories), or 5 μg/ml Blasticidin (Melford Laboratories) and 20 μg/ml G-418. All cell lines generated in this study are available upon request. Sequences of primers used in this study are shown in Supplementary Data 4.

## Generation of KIAP1−3 add back cell lines

To produce the add back cell lines, the *KIAP1* (*LmxM.01.0010*), *KIAP2* (*LmxM.30.0390*) and *KIAP3* (*LmxM.20.1190*) genes were PCR amplified and then cloned into the *HindIII* and *SpeI* or *SpeI* and *BamHI* restriction sites of the constitutive expression plasmid (pJ1364) to generate a C-terminally tagged version of KIAP1 and KIAP2 with an mNG and a triple myc tag or an N-terminally tagged version of KIAP3 with a triple myc tag, respectively, as described previously[43,44]. The plasmids were linearised by digestion with PacI (New England Biolabs, Ipswich, UK) and ethanol precipitated before transfection. The constructs were transfected using a Nucleofector 2b. Successful transfectants were selected with 25 μg/ml phleomycin (Melford Laboratories).

## Light microscopy

For light microscopy of living cells, in vitro haptomonad-like promastigotes attached to a piece of a gridded glass coverslip were washed twice in DMEM (Gibco, UK), incubated in DMEM with 1 μg/ml Hoechst 33342 (Sigma-Aldrich, UK) for 5 min and then washed twice in DMEM. Coverslip pieces were mounted onto another glass coverslip and then onto a glass slide, with the cell attachment side facing up. Attached cells were imaged using a Zeiss Imager Z2 microscope (Carl Zeiss, Jena, Germany) with a Plan-Apochromat 63×/1.4NA Oil objective and a Hamamatsu Flash 4 camera (Hamamatsu Photonics, Hamamatsu, Japan).

## Bioinformatic analysis

The 18S rRNA sequences of the selected kinetoplastids were retrieved from TriTrypDB. Alignment and phylogenetic reconstructions of the 18S rRNA sequences of the kinetoplastids were performed using the function 'build' of ETE3 3.1.2[45] as implemented on the GenomeNet (https://www.genome.jp/tools/ete/). Alignment was performed with MAFFT v6.861b with the default options[46]. Columns with more than 1% of gaps were removed from the alignment using trimAl v1.4.rev6[47]. ML tree was inferred using RAxML v8.2.11 ran with model GTRGAMA and default parameters[48]. Branch supports were computed out of 100 bootstrapped trees.

Reciprocal best BLAST was used to find orthologous sequences, with a minimum *e-value* cutoff of $10^{-5,49}$. *L. mexicana* KIAP protein sequences were used to interrogate the protein sequence database on TriTrypDB by BLASTP, which identified sequences in other kinetoplastid species. The top hit from each species was then used to interrogate the *L. mexicana* genome by BLASTP and if this returned the same *L. mexicana* KIAP protein as used in the initial search the two proteins were considered orthologous. KIAP sequences were analysed by the InterPro (https://www.ebi.ac.uk/interpro/). Protein structures were predicted using AlphaFold[50–52], using the exact pipeline previously described[50]. Visualisation and superposition of protein structures were carried out with PyMOL (DeLano Scientific LLC, San Carlos, CA, http://www.pymol.org). The structure of human Tetranectin was obtained from the RCSB protein data bank (https://www.rcsb.org/; PDB: 1HTN)[21].

## Time-lapse observation

For time-lapse observation, log phase promastigotes ($1 \times 10^6$ cells/ml) were cultured in a μ-dish 35 mm, high glass bottom (iBidi, Gräfelfing, Germany) for 12 h at 28 °C with 5% $CO_2$, and the dish was washed five times with fresh M199 medium before starting imaging. Cells about to adhere to the glass were recorded using a Zeiss LSM 880 confocal microscope (Carl Zeiss, Jena, Germany) with a Plan-Apochromat 63×/1.4NA Oil objective and 488 and 561 nm lasers (1% laser powers) at 28 °C with 5% $CO_2$ in a chamber with controlled temperature and $CO_2$ concentration.

## Confocal microscopy

Log phase promastigotes ($1 \times 10^6$ cells/ml) were cultured in a μ-dish 35 mm, high glass bottom for 24 h, and the dish was washed more than five times with fresh M199 medium to remove as many non-adherent cells as possible before starting imaging. High-resolution confocal microscopy images were acquired with a Zeiss LSM880 with Airyscan using a Plan-Apochromat 100×/1.46NA Oil DIC objective, with 488 and 561 nm lasers. Confocal z-stacks were acquired in superresolution mode using line scanning and the following settings: $40 \times 40$ nm pixel size, 50 nm z-step, 0.66 μs/pixel dwell time, 800 gain and 2% laser power. The z-stack images were processed and analysed using Zen Black software (Carl Zeiss, Jena, Germany) and Fiji[53].

## Analysis of serial section electron microscopy tomography data sets

The tomography data sets that we deposited on the Electron Microscopy Image Archive (EMPIAR; https://www.ebi.ac.uk/empiar/)[54] in our previous study (EMPIAR-11467 and EMPIAR-11468)[13] were re-used for the 3D structural analysis of the attachments of the in vitro haptomonad-like promastigote and in vivo haptomonad promastigote. Three-dimensional models of the electron tomography data were created using 3dmod (IMOD software package, v4.11)[55] as previously described[13].

## Fluorescence recovery after photobleaching

Log phase promastigotes (KIAP1::mNG, SMP1::mCh, $1 \times 10^6$ cells/ml) were cultured in a μ-dish 35 mm, high glass bottom for 24 h, and the dish was washed more than five times with fresh M199 medium to remove as many non-adherent cells as possible before starting imaging. A Zeiss LSM880 with Airyscan was used for imaging and simultaneous photobleaching. The pre-bleach image of the attached flagellum was acquired using a Plan-Apochromat 100×/1.46NA Oil DIC objective, with 488 and 561 nm lasers. An ROI including part of KIAP1::mNG or SMP1::mCh fluorescence signal on a glass surface was imaged five times (short-time observation) or twice (long-time observation) before bleaching. KIAP1::mNG or SMP1::mCh was bleached using the 488 or 561 nm laser, respectively (two or ten iterations of full laser power for the 488 or 561 nm laser, respectively, fluorescence reduction of ~60%). Fluorescence recovery in the KIAP1::mNG or SMP1::mCh was assessed by imaging once every 0.24 s for up to 9.84 s (short-time observation) or every 20 s for up to 300 s (long-time observation) using the 488 and 561 nm low power laser excitation (0.5%). Image analysis was performed using Zen Black software and Fiji. Normalised fluorescence values were calculated by dividing fluorescence value at each time point by average fluorescence value before bleaching and plotted with Microsoft Excel.

## In vitro adhesion quantification

Log phase promastigotes ($5 \times 10^6$ cells/ml) of parental (SMP1::mCh), KIAP1, 2 and 3 knockout and add back cell lines were cultured on ~5 × 5 mm pieces of gridded glass coverslips grid-500 (iBidi) in a 24 well plate with 1 ml of complete M199 medium for 24 h at 28 °C with 5% $CO_2$. The coverslips were washed twice with 1 ml of DMEM, incubated in 1 ml of DMEM with Hoechst 33342 (1 μg/ml) for 5 min, and washed twice with 1 ml of DMEM. The glass pieces were mounted with another glass coverslip on a glass slide. Cells attached in one grid area of 500 μm × 500 μm were imaged using a Zeiss Imager Z2 microscope with a Plan-Apochromat 20×/0.8NA objective and a Hamamatsu Flash 4 camera. The number of cells attached in one grid area was measured based on the phase or bright field image and Hoechst signal using Fiji.

## Sand fly infection experiments

Females of *Lutzomyia longipalpis* were fed through a chick-skin membrane on heat-inactivated sheep blood containing *Leishmania mexicana* promastigotes (Parental (SMP1::mCh), KIAP1, 2 and 3 null mutants and add back cell lines expressing SMP1::mCh, and cell lines expressing fluorescent protein-tagged KIAPs from log-phase cultures (day 3–4) at a concentration $1 \times 10^6$ cells/ml. Blood-engorged females were separated and maintained at 26 °C with free access to 50% sugar solution. On day 6–9 post-blood meal (PBM) females were dissected in drops of saline solution. The individual guts were checked for presence and localisation of *Leishmania*, as well as their morphology under a light and fluorescence microscope Olympus BX51 or Leica SP8 confocal microscope. Special emphasis was given to the colonisation of the stomodeal valve. Levels of *Leishmania* infections were graded into four categories: negative, light (<100 parasites/gut), moderate (100–1000 parasites/gut) and heavy (>1000 parasites/gut). The quantification of infected parasites and localisation in the sand fly was repeated twice.

## Image analysis of movies capturing the sand fly midgut infected with *Leishmania*

Movies capturing the *Leishmania* dynamics in the infected sand fly midguts were recorded using a light microscope Olympus BX53 with an Olympus DP70 camera. Movies were recorded for a duration of 5 seconds at 8 frames per second. To enhance the contrast of moving *Leishmania* cells, image subtraction was performed between two consecutive frames using the 'Image Calculator' function of Fiji[56]. This process generated a series of images, each emphasising the motility of *Leishmania* cells. The resultant images from the subtraction process were subjected to further analysis utilising the 'Temporal-Colour Code' function of Fiji. This function facilitated the representation of cell track by mapping them onto a colour spectrum, thereby providing enhanced visualisation of cell motility within or around the midgut and PSG.

## Metacyclic quantification

On day 9 post blood meal sample sand flies were dissected, and digestive tracts were examined by a light microscope Olympus BX51. Infections were quantitatively assessed by transferring each gut into 150 μl of 0.01% formaldehyde solution, followed by homogenisation and counting using a Burker chamber. *Leishmania* with flagellar length <2 times body length were scored as non-metacyclic forms and those with flagellar length ≥ 2 times body length as metacyclic form[24].

## Calcofluor assay

Midguts dissected in saline were incubated with Calcofluor White Stain (Sigma-Aldrich) which specifically binds to chitin. After 1 min incubation the calcofluor was replaced with the saline solution. The samples were examined under UV light on an Olympus BX51 microscope.

## Immunofluorescence microscopy

Log phase promastigotes ($1 \times 10^6$ cells/ml) of KIAP3 add back cell line which expresses 3Myc::KIAP3 were cultured on 18 × 18 mm precision cover glasses thickness No. 1.5H (Paul Marienfeld, Lauda-Königshofen, Germany) which were sterilised with 100% ethanol in a 6 well plate with 3 ml of complete M199 medium for 24 h at 28 °C with 5% $CO_2$. The coverslips were washed with 3 ml of DMEM, treated with 1% IGEPAL in PEME for 1 min and fixed for 30 min with 4% (w/v)

paraformaldehyde in PEME, washed with phosphate-buffered solution (PBS) three times for 5 min, blocked for 30 min with 1% BSA in PBS at RT. The coverslips were then incubated with c-Myc monoclonal antibody (9E10; Invitrogen, Paisley, UK) in PBS containing 1% BSA (1:100 dilution) overnight at 4 °C, washed with PBS three times for 5 min, incubated with Alexa Fluor 546-conjugated goat anti-mouse secondary antibody (Invitrogen) in PBS containing 1% BSA (1:1000 dilution) for 1 h at RT and washed with PBS three times for 5 min. Then, the coverslips were treated with 0.1 µg/ml Hoechst 33342 in PBS for 5 min, washed with PBS twice and mounted with VECTASHIELD mounting medium (Vector Laboratories, UK) for imaging using a Zeiss Imager Z2 microscope with a Plan-Apochromat 63×/1.4NA Oil objective and a Hamamatsu Flash 4 camera.

### Morphological analysis of in vitro and in vivo *L. mexicana* promastigotes

For morphological analysis of in vitro *L. mexicana* cells, late-log phase promastigotes (~3.7 × 10⁷ cells/ml) were washed twice in DMEM, incubated in DMEM with Hoechst 33342 (1 µg/ml) for 5 min and then washed twice in DMEM. Cells were imaged using a Zeiss Imager Z2 microscope with a Plan-Apochromat 63×/1.4NA Oil objective and a Hamamatsu Flash 4 camera. For morphological analysis of in vivo *L. mexicana* cells, gut smears of *L. mexicana*-infected females on day 9 post-blood meal were fixed with methanol, stained with Giemsa and examined under an Olympus BX51 microscope with an oil-immersion objective. Five morphological forms were distinguished, based on the criteria[23,57,58]: (i) procyclic promastigote: body length <14 µm and flagellar length <body length; (ii) nectomonad promastigote: body length ≥14 µm; (iii) leptomonad promastigote: body length <14 µm and flagellar length <2 times body length; (iv) metacyclic promastigote: body length <14 µm and flagellar length ≥2 times body length, and (v) haptomonad promastigote: characteried by a reduced flagellum with an enlarged flagellar tip.

### Statistical analysis

Means, SDs and SEMs were calculated using Microsoft Excel. For quantification of the number of in vitro attached haptomonad-like promastigotes and midgut width, statistical significance was determined using two-tailed Welch's *t*-test carried out with Microsoft Excel. For quantification of metacyclic cells and the number of cells in the midgut, statistical significance was determined using the Kruskal-Wallis test in Python with Scipy Stats[59]. Differences were considered significant at the level of $p < 0.05$. Data were plotted with Microsoft Excel or the Matplotlib package in Python[60].

### Reporting summary

Further information on research design is available in the Nature Portfolio Reporting Summary linked to this article.

## Data availability

The mass spectrometry proteomics data sets generated in this study have been deposited in the MassIVE under accession code MSV000092919 [https://massive.ucsd.edu/ProteoSAFe/private-dataset.jsp?task=0e17d69a63e44c9b8cf0cc00ebe2ab37] and the ProteomeXchange Consortium under accession code PXD045552. The tomography data sets generated in our previous study[13] have been deposited in the EMPIAR under accession code EMPIAR-11467 and EMPIAR-11468. Source data are provided with this paper.

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

## Acknowledgements

LC-MS/MS was performed using equipment at the University of York Centre of Excellence in Mass Spectrometry, created thanks to a major capital investment through Science City York, supported by Yorkshire Forward with funds from the Northern Way Initiative, and subsequent support from the Engineering and Physical Sciences Research Council (EP/K039660/1; EP/M028127/1). We thank Dr Richard Wheeler (University of Oxford) for the support in the analysis using the AlphaFold and providing the sand fly cartoon in Fig. 4c; the Oxford Brookes Centre for Bioimaging for the support in light and electron microscopy; Prof Eva Gluenz (University of Bern) for providing the *Leishmania major* cell line; Prof Keith Gull (University of Oxford), Prof Sue Vaughan (Oxford Brookes University) and members of the Sunter and Vaughan laboratory for discussions and comments on the manuscript. R.Y. was supported by a JSPS Overseas Research Fellowship and NIBB Collaborative Research Programme (20-515). This work was supported by the Infravec consortium as part of the ISIDORe project (funding from the European Union's Horizon Europe Research & Innovation programme, grant agreement N° 101046133). Work in the lab of JDS is supported by the Wellcome Trust (221944/Z/20/Z). J.S., B.V., K.P. and P.V. were supported by ERD funds, project CeRaViP (16_019/0000759) and Czech Science Foundation (GACR 21-15700S).

## Author contributions

Conceptualisation, R.Y., J.D.S.; Formal Analysis, R.Y., K.P., Funding, R.Y., P.V., J.D.S.; Investigation, R.Y., K.P., B.O.O., E.R.,. F.M.L., A.T., S.N., J.S., B.V.; Supervision, S.N., P.V., J.D.S.; Visualisation, R.Y., K.P.; Writing, R.Y., K.P., J.S., P.V., and J.D.S.

## Competing interests

The authors declare no competing interests.
