## [Peer Review File · Nature Communications]

Discovery of essential kinetoplastid-insect adhesion proteins and their function in Leishmania-sand fly interactionsREVIEWER COMMENTS

Reviewer #1 (Remarks to the Author):

The manuscript Yanase et al. presents an interesting study with novel proteins that may be of relevance to leishmaniasis transmission. The topic is of interest to many in the field of vector and Leishmania biology. It describes a relatively uncharacterised parasite stage that is believed to degrade stomodeal valve in sand flies. There are issues with interpretation and detail that need to be addressed.

1. There is not a clear enough distinction between mutants that cannot colonise sand flies (Beneke 2019) and the KIAP 1 and 3 mutants (fig 4). Flagellar mutants can display similar morphologies and infection outcomes as these. Lower parasite concentrations in infection do not colonise sand fly stomodial valves at same efficiency as higher ones. Low infection and inefficient attachment are not distinguished.
2. The haptomonad stage is not well defined in this study, with too little to distinguish it from dying cells. The KIAP proteins cannot be designated as haptomonad proteins without confirmation of the life stage. Many of the different promastigote stages can attach to slides or sand fly epithelium and this should be discussed (Serafim 2018, Rogers 2009). Flagellar images resemble those from other stages and mutants (Rogers 2008, McCoy 2023).
3. Fluorescent data is overinterpreted in places. Specific functional claims need to be clarified versus speculation throughout. Descriptions ordering attachment plaque construction are somewhat speculative (fig 2). Vesicle transport of flagellar KIAP1 and 3 stretch interpretation beyond presented data.
4. Different stages of promastigotes are captured in images and these need to be labelled. All cells are labelled 'promastigote' except haptomonad, which is also a promastigote stage. This mistaken labeling is misleading and needs correction.

Reviewer #2 (Remarks to the Author):

Leishmania are parasitic protists that cause devastating diseases in tropical and subtropical regions of the globe, representing a major global health burden. These parasites are transmitted by a bite of an infected sand fly. There are 5 developmental stages of the parasite within the insect vector, one of them being the haptomonads that attach to the stomodeal valve in the mouthparts of the sand fly, an organ that mediates uptake of the blood meal into the insect digestive tract. Colonization of the stomodeal valve by Leishmania parasites results in destruction of this structure, causing the fly to make multiple efforts to ingest host blood and resulting in increased regurgitation of parasites into the host tissues. Hence, parasite attachment to this valve is thought to play an important role in transmission. Attachment of the parasite to the stomodeal valve occurs via a unique structure at the tip of the shortened flagellum of the haptomonads, the attachment plaque.

In this paper, the authors have identified the first molecular components of the attachment plaque (AP), three proteins that are called KIAP1, KIAP2, and KIAP3. These proteins were identified using a novel and clever in vitro system that mimics attachment of haptomonads to the stomodeal valve, the attachment of promastigote forms of the parasite to either abraded plastic coverslips or gridded glass coverslips. This model system allowed the authors to identify candidate proteins for the AP by performing comparative proteomics of in vitro attached parasites to non-attached promastigotes. Following filtering of the initial candidates, KIAPs1-3 were identified, by tagging and fluorescent microscopy, as proteins that localized selectively to the tips of short haptomonad flagella where the AP resides. KIAP1 is a type I transmembrane protein with a domain homologous to lectins and is potentially directly involved in parasite attachment, whereas KIAP2 and KIAP3 are not membrane proteins, but both contain a domain with homology to calpains. These proteins also concentrate at deformations in the flagella of promastigotes that are transitioning into haptomonads, and this nascent AP mediates parasite attachment to the substratum from an early time in this transformation.

Perhaps the most important observation is that deletion of the genes for any of these 3 proteins results in inability of parasites to attach to cover slips in vitro or to the stomodeal valve during infections of the sand fly. These are highly significant observations, as they confirm the identification of the very first molecular components of this mysterious parasite structure and they establish the essential nature of the AP in attachment of haptomonads to the insect vector. The study is very thorough, with multiple sophisticated experimental approaches being undertaken to identify these 3 novel proteins and their functions in attachment: development of a revolutionary new in vitro system for haptomonads, quantitative proteomics, high resolution fluorescence microscopy, electron tomography, and characterization of gene knockouts in vitro and in vivo. I believe this will be a high impact study of interest to parasitologists and more broadly to the cell biology community.

I have only a few comments regarding the details of the manuscript.

1. On line 42, the authors should make clear that they are studying the in vitro haptomonads for their proteomic experiments.
2. The UniProt gene accession numbers of KIAP1-3 should be added in the paragraph that starts with line 52 and the subsequent paragraph.
3. The word 'were' should be removed from line 132.
4. One issue that seems to still be unaddressed is whether attachment of parasites to the stomodeal valve is critical for robust transmission to the mammalian host (although this is known for some other parasites, lines 205-207). The authors have an ideal system with which to address this question. While such transmission studies may not be necessary for this study, the authors might reference any publications in the literature that make suggestions one way or the other or that indicate that this question is not resolved. The role of the AP in transmission will certainly be an important issue for future investigations.

Reviewer #3 (Remarks to the Author):

In the manuscript submitted to Nature Communications (id nº NCOMMS-23-46078-T), entitled “Discovery of essential kinetoplastid-insect adhesion proteins and their function in Leishmania-sand fly interactions”, by Yanase et al. the authors propose to dissect the mechanism of adhesion of Leishmania haptomonads to the sandfly stomodeal valve. These are the least study parasite forms in the vector; therefore, this study is a needed addition to the literature. Overall, the manuscript reads well, and the conclusions are supported by the data. However, there are issues that must be addressed before this paper is ready for publication. Additionally, I think the authors can extend the data to address the biological relevance of haptomonad adhesion *in vivo*; this would definitely be a significant piece of information to direct future studies on the matter. Below, I list my reservations in a point-by-point manner.

Major points:

1- The data referring to the proteomics analysis is not clear, in my opinion:

a. You compared attached parasites with parasites in suspension. While I do not think this is a huge limitation, I leave here a question you can then use in your discussion: shouldn't the most appropriate control be the parasites that were subjected to the attachment process but did not attach?

b. You mention you found “371 proteins enriched in the attached haptomonad flagellum”. However, looking at Supplementary table 1, I cannot understand which are these and what were the parameters considered. Looking at the Fischer's exact test, you only have around 130 significant differences (which include proteins enriched in promastigotes versus haptomonads). Looking at the number of proteins absent in the flagella of promastigotes versus haptomonads, I also count around 130. Please explain in detail your exclusion criteria in the manuscript. Additionally, please include a descriptive legend on the data table to allow the readership a fair analysis.

c. You mention you selected 20 candidate proteins based on a genome-wide localization resource. However, in supplementary table 2 you only list the 20 candidate proteins. Please update the table with all 371 proteins subjected to these analysis and explain why the remaining 351 were excluded (and why the 20 were included for the matter). You can add new columns named exclusion criteria (e.g. mitochondrial protein as mentioned in the methods) and inclusion criteria (e.g. I am assuming you selected flagellar proteins). You also need to explain why you chose some with no orthologs in *T. brucei*, and not other (I am assuming you excluded some of these). And finally, you need again to provide an adequate self-explanatory legend for the updated Supplementary Table 2.

2- It would be interesting to see the expression of KIAPS on attached haptomonads *in vivo* (the same super-resolution images would be great, but at least some confocal micrographs). Those images would definitely prove that what you are seeing *in vitro* translates to the *in vivo* context. The KO data are convincing, but these microscopy data would definitely add to the paper.

3- Why is there no data for KIAP2 *in vivo*? If you focus on the 3 adhesion proteins throughout the paper, you also must show these data.

4- Also pertaining to the *in vivo* data, I think an important detail is missing. While there are no parasites colonizing the stomodeal valve, the infection burden is not affected. Therefore, a few questions can be made. Are the parasite plugs formed by KO parasites *in vivo* similar to those formed by WT or AB parasites? Can sandflies infected with these KIAP KO parasites still transmit

Leishmania parasites? If yes, do they do it as efficiently? I think to include these data in the paper is essential. It would add to the discussion of whether the loss of integrity of the stomodeal valve is essential for effective parasite transmission or not. Of note, in my opinion, the inclusion of these data is very important!

5- Finally, in my opinion, there is something that must be discussed. Adhesion of Leishmania parasites to the sandfly gut is not only important in the context of haptomonads, but also of nectomonads. However, as the infection data suggests, while these KIAPs seem to be essential for haptomonad adhesion, they seem not to interfere to the attachment of parasites to the midgut that prevents their excretion together with the bloodmeal remnants – at least in your in vivo context using permissive sandflies. I think you need to discuss this fact and speculate why these 2 processes are different. On this I leave a few open questions. Would you expect a different context considering restrictive vector-parasite pairings? Is the (e.g. sugar) composition of the promastigote flagellum different in KO versus WT parasites? Are the two adhesion processes expected to be equally dynamic?

Minor points:

1- Line 22: the authors state that we can find 5 different Leishmania developmental stages within the sandfly. I count seven: amastigotes, procyclics, nectomonads, leptomonads, haptomonads, retroleptomonads, and metacyclics.

2- Throughout the manuscript the authors do not distinguish axenic from “real” haptomonads. As far as I am concerned, there is only structural data that shows these forms are similar. Do you have any RNA-seq (or other omics) data that can show these in vitro and in vivo forms are similar? If you have, please share it. Otherwise, please clarify throughout the manuscript whether you worked with real haptomonads, or with haptomonad-like cells (as you call them in the previous manuscript).

3- Do you have any explanation to the fact that the KIAP3 addback did not completely recover the adhesion phenotype in vitro?

4- The decreased in vitro adhesion phenotype for KO parasites in vitro is clear. However, you still have some parasites adhering. First, how do you explain this? Second, can you trust the localization data shown in the context of KO parasites?

5- I would like to see KIAP localization data in the context of adhered parasites of a different Leishmania species, as a way to (semi-)validate the panel in Figure 1d.

6- There is an odd mention to proteophosphoglycan secretion by haptomonads. Are you saying that this parasite form could be important for the plug-formation process? Considering the attachment plaque, do you expect secretion to happen equally in these parasite stage? Is the flagellar pocket structure retained unaltered in this parasite stage?

Response to reviewers

Reviewer #1:

1. There is not a clear enough distinction between mutants that cannot colonise sand flies (Beneke 2019) and the KIAP 1 and 3 mutants (fig 4). Flagellar mutants can display similar morphologies and infection outcomes as these. Lower parasite concentrations in infection do not colonise sand fly stomodial valves at same efficiency as higher ones. Low infection and inefficient attachment are not distinguished.

It is first important to understand a fundamental difference between flagellar mutants and our KIAP mutants. The former are mutations in flagellar function itself and exhibit a phenotype *in vitro* in promastigote forms and indeed in all forms where a motile flagellum is required i.e. they have easily observable deficiencies in movement *in vitro*, which translates to similar deficiencies *in vivo* because the mutations are in proteins that have an implicit role in flagellar function. The KIAP mutants are very different. They exhibit no adverse phenotype in promastigote cells and indeed, their expression is focussed to enable the novel function of attachment. The mutants studied by Beneke et al., 2019 that did not colonise the sand flies were those that were either paralysed or swam in an uncoordinated manner. These mutants were unable to migrate to the anterior regions of the midgut (thoracic midgut, cardia) and remained confined to abdominal midgut (see Figure 7 in Beneke et al., 2019). In contrast, our KIAP null mutants migrated to these anterior regions of the midgut where the stomodeal valve is located (Fig. 4c). Moreover, the null mutants were present in equivalent numbers to parental line, see Figure 4b (intensities of infections). The lack of adhesion to the stomodeal valve was therefore caused by the inability of the KIAP mutants to attach and not by lower parasite numbers.

2. The haptomonad stage is not well defined in this study, with too little to distinguish it from dying cells. The KIAP proteins cannot be designated as haptomonad proteins without confirmation of the life stage. Many of the different promastigote stages can attach to slides or sand fly epithelium and this should be discussed (Serafim 2018, Rogers 2009). Flagellar images resemble those from other stages and mutants (Rogers 2008, McCoy 2023).

The *Leishmania* haptomonad form is defined as a cell with a shorter and wider cell body that is attached to the cuticle lining of the stomodeal valve or pylorus region of the hindgut. The adhesion occurs through a highly modified flagellum that is much-reduced in length and into which an organised cytoskeletal structure has been assembled (Killick-Kendrick et al., 1974). The defining feature of the haptomonad form is the strong and stable connection through a modified flagellum to the cuticle surface of the stomodeal valve/pylorus. To demonstrate the KIAPs are bona fide haptomonad proteins we have determined the position and nature of the parasites expressing fluorescent protein-tagged KIAPs in infected sand flies (Fig. 4a and Supplementary Fig. 6a). From these images, it is clear that the parasites expressing these KIAPs are lining the surface of the stomodeal valve exactly as the haptomonad form is defined and that the KIAPs are therefore defined as haptomonad expressed proteins.

We have previously imaged the *in vitro* differentiation of *in vitro* promastigote forms to an *in vitro* attached haptomonad-like form, which occurs through a series of defined stages, using a combination of light and electron microscopy (Yanase et al., 2023). As such, we are experienced in applying criteria to cell type analysis. In this work, we showed that the attachment plaque formed by the *in vitro* attached haptomonad-like form was comparable to that observed in haptomonads attached to the stomodeal valve in the sand fly. We acknowledge our *in vitro* differentiation system will not recapitulate the situation in the sand fly completely and have renamed the haptomonad stage we describe here to the *in vitro* haptomonad-like form. Given our experience of this differentiation system, we are confident the cells we are imaging and defining as *in vitro* haptomonad-like forms are healthy. First, our microscopy shows phase bright cells. Second, as an experimental demonstration, we attempted to stain the *in vitro* haptomonad-like forms with DAPI, a DNA stain that is unable to cross the cell membrane of living cells (see images below). From these data, we were able to show that the *in vitro* haptomonad-like forms are only stained with DAPI after they have been fixed and killed with methanol, again demonstrating these cells are healthy and physiologically intact. Moreover, in our experience, a dying promastigote cell does not disassemble its flagellum before cell death so would be unlikely to look like these cells.

In the sand fly there are two distinct modes of parasite adhesion to the midgut –

- Firstly, the flagellum of the nectomonad form interdigitates its flagellum between the midgut microvilli, with no modification to the ultrastructure of the flagellum (Killick-Kendrick et al., 1974). On the molecular level this midgut attachment has been attributed either to terminal galactose on LPG and sand fly galectin in the specific sand fly vector *P. papatasi* (Kamhawi et al., 2004) or due to the parasite surface and midgut mucin in permissive sand fly species (Myšková et al., 2016) and this interaction is readily reversible.
- Secondly, the highly modified flagellum of the haptomonad form attaches the parasite to the chitin lining of stomodeal valve or pylorus region of the hindgut. This is a strong and stable attachment that is not readily reversed. It is this second, highly structured and cryptic form of attachment that we are studying.

Moreover, when many of the different promastigote stages, with an extended flagellum, are settled on glass they are able to attach but critically this attachment is distinct to that of the haptomonad as the interaction is transient and there is no modification to the flagellum structure. We have previously shown (Yanase et al., 2023) that the *in vitro* haptomonad-like

form we are using here most closely resembles the haptomonad form attached to the stomodeal valve in the sand fly. We agree with the reviewer it is important to consider all modes of *Leishmania* adhesion, and to aid readers and focus on the cryptic and critical stage that we are studying we have now included a discussion of the different modes of *Leishmania* adhesion in the sand fly to our manuscript (Line 392-417).

We have analysed the mutants in the papers the reviewer highlighted. The mutants described in McCoy et al., 2023 are stable null mutants in which the flagellum is unable to assemble in the promastigote form and hence appear with a short stub of flagellum protruding from the cell body. This is therefore very distinct and easily distinguishable from the *in vitro* haptomonad-like cells in which we are able to follow their differentiation, see that they have a long flagellum and actually watch the disassembly of the flagellum to generate attached cells. These may appear superficially similar to those with a defect in flagellum assembly but their biology is totally different and have in fact assembled an attachment plaque creating a strong attachment to the glass.

We know of only one paper, Rogers et al., 2008. In this paper, they described the effect of chitinase overexpressing mutant on the sand fly and onward transmission; however, they show no light micrographs of the mutant. The only images presented were electron micrographs of haptomonad forms attached to the stomodeal valve through a modified flagellum, showing this mutant is able to attach to the valve. We have now referred to this paper in our discussion where we consider the function of the haptomonad form and its role in damaging the cuticle of the stomodeal valve (Line 418-440).

3. Fluorescent data is overinterpreted in places. Specific functional claims need to be clarified versus speculation throughout. Descriptions ordering attachment plaque construction are somewhat speculative (fig 2). Vesicle transport of flagellar KIAP1 and 3 stretch interpretation beyond presented data.

We apologise for the lack of distinction between our results and elements of discussion. We have re-written sections of the manuscript and moved all discussion and speculation to a separate discussion section. We have found that the mechanism of attachment plaque construction is of great interest to people in the field and those more widely interested in cell-to-cell and cell-to-substrate attachment in many other systems. Therefore, we felt it is important and will assist to give some thoughts on this in the discussion. We have therefore improved these descriptions regarding attachment plaque construction and focussed them in the discussion, with an acknowledgment they are informed speculations (Line 377-391). We agree that the discussion points about vesicle transport in the flagellum are too speculative and have removed these from the revised version of the manuscript.

4. Different stages of promastigotes are captured in images and these need to be labelled. All cells are labelled 'promastigote' except haptomonad, which is also a promastigote stage. This mistaken labeling is misleading and needs correction.

We agree with the reviewer that based on the Hoare and Wallace, 1966 definition for a promastigote (a form with the kinetoplast anterior to the nucleus and a flagellum arising near and emerging from the anterior end of the cell body) that all our images show promastigote forms. However, subsequent analysis after that date defined a number of different promastigote stages during a sand fly infection, including for example metacyclic promastigotes and

leptomonad promastigotes. These definitions are based on morphological characteristics and there are few (if any) specific and reliable molecular markers that we can apply to discriminate between these forms; without such markers we would prefer not to assign a specific stage to the cells we have imaged, apart from the haptomonad form, which is clearly distinct as it is attached and non-motile, and for which we have now identified specific marker proteins.

Moreover, whilst our work (and from others in the field) uses *Leishmania mexicana* parasites that retain infectivity in both host and vector (Corrales et al., 2021; Sunter et al., 2019); it is still not clear how relevant the morphological defined forms identified in the sand fly are to the culture parasite in the literature. It has become standard practice to refer to the culture form of the insect stages of the parasite as simply promastigotes (Geoghegan et al., 2022; Gluenz et al., 2010). In this revision, we have specifically defined all *in vitro* non-attached cells as *in vitro* promastigote cells and the *in vitro* derived haptomonad-like cells as *in vitro* haptomonad-like cells to clearly differentiate them from the stages in the *in vivo* environment of the sand fly. We have changed the labelling to reflect this.

Reviewer #2:

1. On line 42, the authors should make clear that they are studying the *in vitro* haptomonads for their proteomic experiments.

We appreciate the suggestion and have decided to refer to *in vitro* induced attached cells as *in vitro* haptomonad-like cells throughout this paper, as we referred to in our previous paper (Yanase et al., 2023), to make clear at which analyses were done with *in vitro* derived haptomonads.

2. The TriTryp gene accession numbers of KIAP1-3 should be added in the paragraph that starts with line 52 and the subsequent paragraph.

We have added the TriTrypDB gene accession numbers of KIAP1-3 in the corresponding paragraphs (Line 75-81).

3. The word ‘were’ should be removed from line 132.

We have removed the word ‘were’.

4. One issue that seems to still be unaddressed is whether attachment of parasites to the stomodeal valve is critical for robust transmission to the mammalian host (although this is known for some other parasites, lines 205-207). The authors have an ideal system with which to address this question. While such transmission studies may not be necessary for this study, the authors might reference any publications in the literature that make suggestions one way or the other or that indicate that this question is not resolved. The role of the AP in transmission will certainly be an important issue for future investigations.

We thank the reviewer for making this important comment and it is something we think about a lot. The complexity of defining experiments in this area is beyond the scope of this study which is already covering a huge amount of ground, from molecular discovery to functionally phenotyping *in vitro* and *in vivo*. However, to provide greater foundational context for these

future transmission experiments in this work, we performed additional experiments to investigate the effect of infection with KIAP null mutants on the sand fly vector (Fig. 5).

Key features of a late sand fly infection are i) the production of the promastigote secretory gel which distends the midgut of the sand fly and ii) the damage to the cuticle surface of the stomodeal valve (Fig. 5).

- i) We examined the structure of the thoracic midgut of sand flies infected with parental, KIAP null mutants, and add backs by light microscopy. We first noted that the midguts infected with the KIAP null mutants were narrower and less distended than those infected with parental and add back parasites. Moreover, when we examined movies of the infected midguts, we saw a dramatic difference in parasite behaviour. In midguts infected with parental and add back parasites, they were unable to move, even when the midgut was ruptured, as the parasites were embedded in the promastigote secretory gel. Conversely, in midguts infected with the KIAP null mutants, the cells were able to move, and when the midgut was ruptured the cells swam freely and only a limited amount of promastigote secretory gel was seen.
- ii) We examined the effect on the chitin of the cuticle surface of the stomodeal valve in sand flies infected with parental, KIAP null mutants, and add backs by staining with calcofluor. In uninfected midguts, there was clear distinct chitin staining from the valve and crop and this was replicated in the KIAP null mutants. However, in the midguts infected with parental and add-back parasites there was a distinct loss of chitin staining, with a deformation of the valve form.

Overall, we now show that the loss of parasite adhesion has a dramatic effect on sand fly physiology, with a reduction in stomodeal valve damage and midgut distortion. Moreover, there is less promastigote secretory gel present, though the percentage of metacyclics remained the same. We have discussed these new data integrating previous relevant work such, as the chitinase overexpression mutant, which enhances transmission (Rogers et al., 2008). We anticipate these effects will likely reduce the chances of transmission to and/or pathology in the mammalian host. Therefore, although we have added foundational data, we agree with the reviewer that dissecting the complex issue of transmission is beyond the scope of this publication.

Reviewer #3:

Major points:

1- The data referring to the proteomics analysis is not clear, in my opinion:

a. You compared attached parasites with parasites in suspension. While I do not think this is a huge limitation, I leave here a question you can then use in your discussion: shouldn't the most appropriate control be the parasites that were subjected to the attachment process but did not attach?

We appreciate the reviewer for the suggestion. We could have used the parasites that were subjected to the adhesion process but did not attach for the proteomic analysis. However, what we wanted to do here was to exclude the components of the non-attached flagellum from the components of the attached flagellum, which contain the attachment plaque structures. In retrospect, performing the proteomic analysis on those cells that did not attach in addition to those that attached and those in suspension would have provided additional informative data

but as the reviewer points out we were still able to identify critical attachment plaque proteins. We have mentioned this point in our discussion (Line 334-342).

b. You mention you found “371 proteins enriched in the attached haptomonad flagellum”. However, looking at Supplementary table 1, I cannot understand which are these and what were the parameters considered. Looking at the Fischer’s exact test, you only have around 130 significant differences (which include proteins enriched in promastigotes versus haptomonads). Looking at the number of proteins absent in the flagella of promastigotes versus haptomonads, I also count around 130. Please explain in detail your exclusion criteria in the manuscript. Additionally, please include a descriptive legend on the data table to allow the readership a fair analysis.

The 371 proteins enriched in the attached haptomonad flagellum are all those proteins for which there are more peptide spectrum counts detected in the attached sample versus the non-attached sample, after removing the non-*Leishmania* proteins and regardless of the Fischer’s exact test value.

From these 371, we considered only those proteins which were at least 8-fold enriched in the attached sample versus the non-attached sample again without considering the Fischer’s exact test value. To this set of 143 proteins, we applied a series of exclusion criteria:

i) proteins with an annotation or protein domain that suggested a function unrelated to attachment e.g. metabolism, mitochondrial, protein folding and assembly, endocytic trafficking, transcription, transporters, proteasome, cell cycle kinase were excluded.

ii) proteins with a known function or localisation e.g. SMP1, intraflagellar transport (IFT) proteins, kinesins were excluded.

iii) pseudogenes and non-scaffolded genes were excluded.

This gave a set of 39 proteins in which were prioritised hypothetical proteins and those with potential cytoskeletal functions, calcium binding and cAMP signalling (Denecke et al., 2022). These 39 were further refined by analysing the annotation and localisation of the orthologs if present in *Trypanosoma brucei*. Proteins whose ortholog localised to the mitochondrion, nucleus, cytoplasm, or endocytic system were excluded. *Leishmania* specific proteins and those whose orthologs had a cytoskeletal localisation or background signal were prioritised to give the set of 20 proteins which were screened in the manuscript.

We have now included an extended description of our filtering process in the manuscript (Line 53-74) and have added an additional spreadsheet to show the intermediate steps, with detailed legends – see Supplementary Data 2 and 3.

c. You mention you selected 20 candidate proteins based on a genome-wide localization resource. However, in supplementary table 2 you only list the 20 candidate proteins. Please update the table with all 371 proteins subjected to these analysis and explain why the remaining 351 were excluded (and why the 20 were included for the matter). You can add new columns named exclusion criteria (e.g. mitochondrial protein as mentioned in the methods) and inclusion criteria (e.g. I am assuming you selected flagellar proteins). You also need to explain why you chose some with no orthologs in *T. brucei*, and not other (I am assuming you excluded some of these). And finally, you need again to provide an adequate self-explanatory legend for the updated Supplementary Table 2.

We have now included an extended description of the pipeline we used to define our candidates that we took forward to the tagging screen. In addition, we have included additional spreadsheets (Supplementary Data 2 and 3), with detailed legends that show more clearly our filtering process.

The refinement of our candidate list using TrypTag and *T. brucei* was the final step. We analysed the 39 proteins which had not been excluded based on their annotation in the previous step by identifying those with orthologs in *T. brucei*. If the protein did not have an ortholog and was *Leishmania* specific the protein was included in the tagging screen. If the protein did have an ortholog and its localisation suggested a cytoskeletal localisation then it was included in the tagging screen. However, if the protein did have an ortholog and its localisation and/or annotation suggested a function unlikely related to adhesion e.g. mitochondrial it was not included.

2- It would be interesting to see the expression of KIAPS on attached haptomonads in vivo (the same super-resolution images would be great, but at least some confocal micrographs). Those images would definitely prove that what you are seeing in vitro translates to the in vivo context. The KO data are convincing, but these microscopy data would definitely add to the paper.

Based on the reviewer's suggestion we infected sand flies with parasites expressing fluorescent protein-tagged KIAPs and imaged the cardia and stomodeal valve region after 8 days post blood meal (Fig. 4a and Supplementary Fig. 6a). These images show that the KIAPs are predominantly expressed in the haptomonad forms that are attached to the stomodeal valve. The signal from the tagged KIAPs clearly outlines the stomodeal valve. This confirms that the KIAPs are expressed in the haptomonads in the sand fly and are markers for this life cycle stage.

3- Why is there no data for KIAP2 in vivo? If you focus on the 3 adhesion proteins throughout the paper, you also must show these data.

We have now examined the function of KIAP2 using the same null mutant and add back approach we used for KIAP1 and KIAP3. These data show that KIAP2 is essential for adhesion to the stomodeal valve in the sand fly and that attachment is recovered when KIAP2 expression is restored (Fig. 4b-d).

4- Also pertaining to the in vivo data, I think an important detail is missing. While there are no parasites colonizing the stomodeal valve, the infection burden is not affected. Therefore, a few questions can be made. Are the parasite plugs formed by KO parasites in vivo similar to those formed by WT or AB parasites? Can sandflies infected with these KIAP KO parasites still transmit *Leishmania* parasites? If yes, do they do it as efficiently? I think to include these data in the paper is essential. It would add to the discussion of whether the loss of integrity of the stomodeal valve is essential for effective parasite transmission or not. Of note, in my opinion, the inclusion of these data is very important!

Reviewer 2 raised a similar question about what effect loss of parasite adhesion to the valve would have on transmission. They acknowledge we have the requisite mutants now to address this complex issue but that this would be beyond the scope of this manuscript. We agree with the reviewers that the issue of transmission is important; however, the focus of this paper was

on dissecting the adhesion mechanism and we have provided important molecular insights into this previously cryptic process. We have added some foundational work to this paper to presage the amount of underpinning data that will be needed in planning definitive future experiments.

We have examined the effect of *Leishmania* infection on the sand fly in more detail (Fig. 5, detailed response above) to provide crucial data that will inform future transmission experiments. Our data show that the KIAP null mutants cause much less damage to the cuticle surface of the valve, with a reduction in the distortion of the midgut. Moreover, the KIAP null mutants move freely within the thoracic midgut as there is less promastigote secretory gel. This sets the scene for the interpretation of transmission experiments.

5- Finally, in my opinion, there is something that must be discussed. Adhesion of *Leishmania* parasites to the sandfly gut is not only important in the context of haptomonads, but also of nectomonads. However, as the infection data suggests, while these KIAPs seem to be essential for haptomonad adhesion, they seem not to interfere to the attachment of parasites to the midgut that prevents their excretion together with the bloodmeal remnants – at least in your in vivo context using permissive sandflies. I think you need to discuss this fact and speculate why these 2 processes are different. On this I leave a few open questions. Would you expect a different context considering restrictive vector-parasite pairings? Is the (e.g. sugar) composition of the promastigote flagellum different in KO versus WT parasites? Are the two adhesion processes expected to be equally dynamic?

Thank you for this suggestion, we have used it to develop a section within the discussion in which we compare and contrast the modes of *Leishmania* adhesion (Line 392-417).

There are two distinct types of *Leishmania* adhesion within the sand fly. Firstly, the flagellum of the nectomonad interdigitates between the midgut microvilli and secondly, the highly modified flagellum of the haptomonad form adheres the parasite to the chitin lining of the stomodeal valve or pylorus region of the hindgut. The adhesion of haptomonads to the hindgut is restricted to *Leishmania* species in the *Viannia* subgenus such as *L. braziliensis*; whereas haptomonad attachment to the stomodeal valve is ubiquitous across human-infective *Leishmania*. The adhesion of the nectomonad form to the midgut microvilli is required to ensure the parasite is not excreted with the blood meal remnants and as the KIAP null mutants were able to migrate to the anterior midgut this adhesion was still occurring. The KIAP null mutants were tested in a permissive sand fly species (*L. longipalpis*) in which the adhesion relies on the interaction between the parasite surface and midgut mucin. There are specific sand fly and *Leishmania* species pairings (e.g. *L. major* and *P. papatasi*) in which the midgut adhesion relies on the LPG terminal galactose and the sand fly galectin. The protein domain predictions for KIAP1-3 do not suggest a role in LPG synthesis, so it is likely that even in restrictive sand flies the KIAPs do not have a role in mediating nectomonad adhesion.

Moreover, the dynamics and ultrastructural changes associated with the two types of *Leishmania* adhesion are distinct, suggesting they occur through independent mechanisms. The interaction between the nectomonad flagellum and the microvilli is not associated with any ultrastructural changes to the flagellum and is reversible enabling the onward migration of the parasite to the anterior midgut; whereas, haptomonad adhesion requires a wholesale remodelling of the flagellum, creating a strong and stable connection to the sand fly. However, despite these distinctions the initial interaction between the *Leishmania* parasite and the stomodeal valve will be mediated by the parasite surface and may have commonalities with nectomonad adhesion. In addition, the capacity of haptomonads to differentiate into leptomonads/metacyclics is unknown but the stability of the attachment plaque suggests that

the attached cell is terminally differentiated. Interestingly, we previously observed that ~1% of haptomonads were dividing (Yanase et al., 2023), which may generate a free-swimming daughter and such asymmetric differentiation divisions are found in other kinetoplastid parasites.

Minor points:

1- Line 22: the authors state that we can find 5 different *Leishmania* developmental stages within the sandfly. I count seven: amastigotes, procyclics, nectomonads, leptomonads, haptomonads, retroleptomonads, and metacyclics.

We have modified this sentence to explain there are 6 developmental stages in the sand fly (Line 28-29). At the moment there is not unequivocal data to show that the retroleptomonad stage has distinctive characteristics and indeed a recent preprint has shown that a second blood meal does not result in the generation of a distinct retroleptomonad stage (Catta-preta et al., 2024). Therefore, we prefer to only refer to the canonical *Leishmania* developmental stages.

2- Throughout the manuscript the authors do not distinguish axenic from “real” haptomonads. As far as I am concerned, there is only structural data that shows these forms are similar. Do you have any RNA-seq (or other omics) data that can show these in vitro and in vivo forms are similar? If you have, please share it. Otherwise, please clarify throughout the manuscript whether you worked with real haptomonads, or with haptomonad-like cells (as you call them in the previous manuscript).

As you suggest, we clarified throughout the manuscript whether we worked with “real” haptomonads, or with haptomonad-like cells.

3- Do you have any explanation to the fact that the KIAP3 addback did not completely recover the adhesion phenotype in vitro?

We do not have a specific explanation but we can speculate. The KIAP3 protein expressed in the add-back cell line is a mutant, as it is fused to the mNG fluorescent protein that was used to confirm expression and correct localisation. The presence of the tag may interfere with KIAP3 function. Moreover, only one copy of KIAP3 was introduced into the add-back cell level and it is possible that the KIAP3 expression is lower than the endogenous in the parental cells.

4- The decreased in vitro adhesion phenotype for KO parasites in vitro is clear. However, you still have some parasites adhering. First, how do you explain this? Second, can you trust the localization data shown in the context of KO parasites?

In the null mutants, only one KIAP is lost and the other components required for haptomonad adhesion are still present. For example, in Figure 3a, you can see that when one of the KIAP proteins is deleted, the other KIAPs are still expressed. It appears that in this situation adhesion can still occur at a very low frequency, facilitated by the remaining KIAPs.

Careful interpretation of the localisation data shown in the context of null mutants is required. We showed in the previous paper that the attachment plaque of the haptomonad is highly

organised structure (Yanase et al., 2023) and this organisation is potentially disrupted in the null mutants. Our data is informative of whether the KIAPs are able to localise to structures within a flagellum that is attempting to attach; however, this is unlikely to reflect the organisation within an adhering parental cell. We have included this point in our discussion (377-391).

5- I would like to see KIAP localization data in the context of adhered parasites of a different *Leishmania* species, as a way to (semi-)validate the panel in Figure 1d.

We have now endogenously tagged KIAP1, KIAP2 and KIAP3 in *Leishmania major*, an old-world species of *Leishmania*. We have imaged these parasites as *in vitro* promastigotes and *in vitro* haptomonad-like cells (Supplementary Fig. 3). The localisation pattern of the KIAPs is similar between *L. mexicana* and *L. major*, with an increased fluorescence signal from the enlarged tip of the attached flagellum, though the amount of enrichment of KIAP1 in *L. major* is lower than in *L. mexicana*.

6- There is an odd mention to proteophosphoglycan secretion by haptomonads. Are you saying that this parasite form could be important for the plug-formation process? Considering the attachment plaque, do you expect secretion to happen equally in these parasite stage? Is the flagellar pocket structure retained unaltered in this parasite stage?

We agree this is a speculative point and was not as clearly explained we intended. Within the haptomonad flagellum in the sand fly we have observed vesicles that are in close proximity and fused to the attachment interface. In these vesicles, we often observed material that had the appearance of the filamentous material surrounding the cells. We therefore speculated that these vesicles may contain fPPG that was being secreted. Given that fPPG is a major component of the promastigote secretory gel and that in the KIAP null mutants there is less gel present it appears there might be a link between the haptomonad and fPPG secretion but to define this connection is beyond the scope of this manuscript. On reflection, we have therefore decided not to discuss the secretion of fPPG, as it is not the main focus of the manuscript.

References

- Beneke T, Demay F, Hookway E, Ashman N, Jeffery H, Smith J, Valli J, Becvar T, Myskova J, Lestinova T, Shafiq S, Sadlova J, Volf P, Wheeler RJ, Gluenz E. 2019. Genetic dissection of a *Leishmania* flagellar proteome demonstrates requirement for directional motility in sand fly infections. *PLoS Pathog* **15**:1–31. DOI:10.1371/journal.ppat.1007828
- Catta-preta C, Ghosh K, Sacks D, Ferreira T. 2024. Single-cell atlas of *Leishmania major* development in the sandfly vector reveals the heterogeneity of transmitted parasites and their role in infection. *Res Sq*. DOI:https://doi.org/10.21203/rs.3.rs-4022188/v1
- Corrales RM, Vaselek S, Neish R, Berry L, Brunet CD, Crobu1 L, Kuk1 N, Mateos-Langerak J, Robinson DR, Volf P, Mottram JC, Sterkers Y, Bastien P. 2021. The kinesin of the flagellum attachment zone in *Leishmania* is required for cell morphogenesis, cell division and virulence in the mammalian host. *PLoS Pathog* **17**:1–29. DOI:10.1371/journal.ppat.1009666

- Denecke S, Malfara MF, Hodges KR, Holmes NA, Williams AR, Julia H, Pascarella JM, Daniels AM, Sterk GJ, Leurs R, Ruthel G, Hoang R, Povelones ML, Povelones M, Clinical C, Office T, Alto P, Amsterdam VU. 2022. Adhesion of *Crithidia fasciculata* promotes a rapid change in developmental fate driven by cAMP signaling. *bioRxiv*. DOI:<https://doi.org/10.1101/2022.10.06.511084>
- Geoghegan V, Carnielli JBT, Jones NG, Saldivia M, Antoniou S, Hughes C, Neish R, Dowle A, Mottram JC. 2022. CLK1/CLK2-driven signalling at the *Leishmania* kinetochore is captured by spatially referenced proximity phosphoproteomics. *Commun Biol* **5**:1–17. DOI:10.1038/s42003-022-04280-1
- Gluezn E, Ginger ML, McKean PG. 2010. Flagellum assembly and function during the *Leishmania* life cycle. *Curr Opin Microbiol* **13**:473–479. DOI:10.1016/j.mib.2010.05.008
- Hoare CA, Wallace FG. 1966. Developmental Stages of Trypanosomatid Flagellates: a New Terminology. *Nature* **212**:1385–1386.
- Kamhawi S, Ramalho-Ortigao M, Van MP, Kumar S, Lawyer PG, Turco SJ, Barillas-Mury C, Sacks DL, Valenzuela JG. 2004. A role for insect galectins in parasite survival. *Cell* **119**:329–341. DOI:10.1016/j.cell.2004.10.009
- Killick-Kendrick R, Molyneux DH, Ashford RW. 1974. *Leishmania* in phlebotomid sandflies I. Modifications of the flagellum associated with attachment to the mid-gut and oesophageal valve of the sandfly. *Proc R Soc B Biol Sci* **187**:409–419. DOI:<https://doi.org/10.1098/rspb.1974.0085>
- McCoy CJ, Paupelin-Vaucelle H, Gorilak P, Beneke T, Varga V, Gluezn E. 2023. ULK4 and Fused/STK36 interact to mediate assembly of a motile flagellum. *Mol Biol Cell* **34**:1–17. DOI:10.1091/mbc.E22-06-0222
- Myšková J, Dostálová A, Pěničková L, Halada P, Bates PA, Volf P. 2016. Characterization of a midgut mucin-like glycoconjugate of *Lutzomyia longipalpis* with a potential role in *Leishmania* attachment. *Parasites and Vectors* **9**:1–10. DOI:10.1186/s13071-016-1695-y
- Rogers ME, Hajmová M, Joshi MB, Sadlova J, Dwyer DM, Volf P, Bates PA. 2008. *Leishmania* chitinase facilitates colonization of sand fly vectors and enhances transmission to mice. *Cell Microbiol* **10**:1363–1372. DOI:10.1111/j.1462-5822.2008.01132.x
- Sunter JD, Yanase R, Wang Z, Catta-Preta CMC, Moreira-Leite F, Myskova J, Pruzinova K, Volf P, Mottram JC, Gull K. 2019. *Leishmania* flagellum attachment zone is critical for flagellar pocket shape, development in the sand fly, and pathogenicity in the host. *Proc Natl Acad Sci U S A* **116**:6351–6360. DOI:10.1073/pnas.1812462116
- Yanase R, Moreira-Leite F, Rea E, Wilburn L, Sadlova J, Vojtkova B, Pruzinová K, Taniguchi A, Nonaka S, Volf P, Sunter JD. 2023. Formation and three-dimensional architecture of *Leishmania* adhesion in the sand fly vector. *Elife* **12**:1–23. DOI:10.7554/eLife.84552

REVIEWER COMMENTS

Reviewer #1 (Remarks to the Author):

The authors have addressed most of the suggestions exceedingly well.

The exception to this is the incongruity between clumping 6 lifecycle stages into the generalised term 'promastigotes' when the manuscript attempts to define novel proteins specific to the relatively unexplored haptomonad promastigote stage. Their argument to dismiss the need to outline the 6 promastigote lifecycle stages is circular and inappropriately self-refers to their own publication and expertise repeatedly.

Historically, those working only in prolonged axenic cultures would simplify the lifecycle to only promastigote:amastigote as their cells were no longer capable of differentiation to the other lifecycle stages; selected for replication efficiency over differentiation capacity. This is why Leishmania researchers track culture passages very carefully and must passage the parasites through host animals periodically. Leishmania mexicana keeps this differentiation capacity longer than most species to around passage 8, making it an advantageous model.

In contrast, researchers examining stages within sand flies describe the distinct promastigote stages. This is illustrated as far back as Killick-Kendrick et al (1974) cited by these authors in their response, that describes and compares electron microscope images of haptomonad flagellar morphology to nectomonad microvilli attachment. There are far more recent, pertinent papers regarding haptomonad stomodial valve attachment that continue to describe this lifecycle stage in proper biological context (Alcolea et al. 2019, DOI: 10.1371/journal.pntd.0007288; Morrison et al. 2012, DOI: 10.1111/j.1462-5822.2012.01798.x). Outlining the distinct promastigote stages is therefore essential to the purpose, impact and interpretation of this paper's outcomes.

Reviewer #2 (Remarks to the Author):

This revised version of the original submission addresses the comments of each reviewer, and in my opinion, the responses are appropriate. In particular, I consider the additional studies on the role of KIAPs in the infected sand fly (Fig. 5), in addition to the studies on chitin degradation by wild type, KIAP null, and add back parasites in Fig. 4d, to be a significant addition to the study. As the authors point out, these results begin to address the question of the role of KIAPs and of haptomonads in both the sand fly and for infection of the mammalian host.

Looking over the paper in detail, I did notice a concern, which I missed in the first review, that I believe it would be desirable to address in the final manuscript. The proteomics analysis is based upon quantitative differences in peptide spectrum counts in samples from attached haptomonads

flagella versus non-attached promastigote flagella. How are these two samples compared quantitatively, for instance to conclude that there is an 8-fold difference between the samples?

One would typically apply some type of normalization especially given that the source of the two samples is quite different. Were the samples derived from the same number of starting parasites of each type, or was there an internal normalization performed? Or was there no normalization, and the authors are simply focusing on peptides that were more abundant in the haptomonad sample regardless of the bulk differences between the two biological preparations? Furthermore, it appears (e.g., Supplementary Dataset 1) that there was only a single sample of each type analyzed rather than the use of replicates that is more commonly employed in proteomic studies.

While the body of work makes it clear that the authors have identified proteins that are specific and functionally important for the haptomonad attachment plaque, it would nonetheless be useful to explain and justify the quantitative proteomics strategy that is the basis for the rest of the study.

Reviewer #3 (Remarks to the Author):

Most of my concerns were properly addressed.

Response to reviewers (2nd revision)

Reviewer #1 (Remarks to the Author):

The authors have addressed most of the suggestions exceedingly well.

We would like to thank the reviewer for their appreciation of our additional work that extended and enhanced the original manuscript.

The exception to this is the incongruity between clumping 6 lifecycle stages into the generalised term 'promastigotes' when the manuscript attempts to define novel proteins specific to the relatively unexplored haptomonad promastigote stage. Their argument to dismiss the need to outline the 6 promastigote lifecycle stages is circular and inappropriately self-refers to their own publication and expertise repeatedly.

Historically, those working only in prolonged axenic cultures would simplify the lifecycle to only promastigote:amastigote as their cells were no longer capable of differentiation to the other lifecycle stages; selected for replication efficiency over differentiation capacity. This is why *Leishmania* researchers track culture passages very carefully and must passage the parasites through host animals periodically. *Leishmania mexicana* keeps this differentiation capacity longer than most species to around passage 8, making it an advantageous model.

In contrast, researchers examining stages within sand flies describe the distinct promastigote stages. This is illustrated as far back as Killick-Kendrick et al (1974) cited by these authors in their response, that describes and compares electron microscope images of haptomonad flagellar morphology to nectomonad microvilli attachment. There are far more recent, pertinent papers regarding haptomonad stomodial valve attachment that continue to describe this lifecycle stage in proper biological context (Alcolea et al. 2019, DOI: 10.1371/journal.pntd.0007288; Morrison et al. 2012, DOI: 10.1111/j.1462-5822.2012.01798.x). Outlining the distinct promastigote stages is therefore essential to the purpose, impact and interpretation of this paper's outcomes.

We appreciate these comments and have now considered the different promastigote stages in our results, using the approach outlined in Morrison *et al.*, 2012¹, as highlighted by the reviewer. We have examined our late stage *in vitro* cultures and identified the different promastigote stages and examined the localisation of the KIAPs in procyclic, nectomonad, leptomonad, and metacyclic cells (Supplementary Fig. 2). The localisation of KIAP1-3 is consistent within all these promastigote stages, but it is only in the attached haptomonad-like cell that we see an increase in signal associated with the attached flagellum. We have now included these data and added a description to the manuscript about the localisation in the different stages (lines 82-101). We believe this both clarifies the nomenclature and provides further precision on KIAP expression and localisation.

In addition, we have further analysed the smears taken from the late stage (day 9 PBM) of the infections of sand fly with the KIAP mutants and addbacks (Supplementary Fig. 7d). For all these cell lines we were able to readily observe nectomonad, leptomonad and metacyclic forms, while haptomonads were only observed with the parental and addback cells and not with the KIAP null mutants. As we only examined late stage infections no procyclic promastigotes were seen.

Reviewer #2 (Remarks to the Author):

This revised version of the original submission addresses the comments of each reviewer, and in my opinion, the responses are appropriate. In particular, I consider the additional studies on the role of KIAPs in the infected sand fly (Fig. 5), in addition to the studies on chitin degradation by wild type, KIAP null, and add back parasites in Fig. 4d, to be a significant addition to the study. As the authors point out, these results begin to address the question of the role of KIAPs and of haptomonads in both the sand fly and for infection of the mammalian host.

We appreciate these comments and support acknowledging the substantial additional work and insight we have provided to this important understudied area of *Leishmania* biology.

Looking over the paper in detail, I did notice a concern, which I missed in the first review, that I believe it would be desirable to address in the final manuscript. The proteomics analysis is based upon quantitative differences in peptide spectrum counts in samples from attached haptomonads flagella versus non-attached promastigote flagella. How are these two samples compared quantitatively, for instance to conclude that there is an 8-fold difference between the samples?

One would typically apply some type of normalization especially given that the source of the two samples is quite different. Were the samples derived from the same number of starting parasites of each type, or was there an internal normalization performed? Or was there no normalization, and the authors are simply focusing on peptides that were more abundant in the haptomonad sample regardless of the bulk differences between the two biological preparations? Furthermore, it appears (e.g., Supplementary Dataset 1) that there was only a single sample of each type analyzed rather than the use of replicates that is more commonly employed in proteomic studies.

While the body of work makes it clear that the authors have identified proteins that are specific and functionally important for the haptomonad attachment plaque, it would nonetheless be useful to explain and justify the quantitative proteomics strategy that is the basis for the rest of the study.

The proteomic analysis was designed as a screen and not a global assessment of the attached flagellum. Thus, for our analysis, we focussed on the differences between the total spectrum counts for the proteins present in the attached and unattached flagellum samples. The total spectrum counts for a protein are a measure of its abundance in that sample and we calculated the fold difference between the protein abundance in the different samples based on the total spectrum counts. This was directly analytical which was very successful and informative as we identified the proteins essential for haptomonad adhesion.

Furthermore, we performed a quality control check on the proteomic dataset before analysing it further. From our SEM images we knew that the axoneme was intact in both samples and therefore there should be limited change in abundance of axonemal components between the attached and unattached proteins samples. We therefore examined the relative change in abundance of 10 axonemal proteins (LmxM.13.0430, LmxM.27.0520, LmxM.08_29.0660, LmxM.20.1400, LmxM.36.6380, LmxM.31.0230, LmxM.10.1190, LmxM.23.1310, LmxM.33.3880, LmxM.24.1030) and found that their change in abundance varied from 0.5-2-fold between the samples. The rationale for the 8-fold difference we chose as our cut-off was

that it was well above this level, indicating proteins that were dramatically more abundant in the attached flagellum sample. Moreover, for 129 out of the 143 proteins, which were above the 8-fold cut-off, peptides were only detected in the attached sample, and KIAP1, KIAP2, and KIAP3 were in this category. We have now included these additional details into our methods.

We would like to emphasise that in this study, we used our comparative proteomic approach as a screen to identify candidate proteins, with a potential role in *Leishmania* adhesion, and as the reviewer pointed out our approach was successful in identifying ‘specific and functionally important’ proteins. At no point do we claim that we have defined the proteome of the attachment plaque and we have now modified the text to make this more explicit and to outline the rationale of using a proteomic screen (lines 306-308). This may be useful for other studies.

Our design of a discovery pipeline relied on an initial proteomic screen followed up by a more detailed characterisation of candidate through tagging and localisation. This second screening step meant that the higher false positive rate that would occur from analysing a single biological repeat by proteomics would not affect the strength and validity of our conclusions from this study. It is clear that this 2-step filtering screen has revealed new protein functions and that the KIAPs are essential for mediating *Leishmania* adhesion to the stomodeal valve of the sand fly.

Reviewer #3 (Remarks to the Author):

Most of my concerns were properly addressed.

We are pleased to see the support of this reviewer and the appreciation of the additional data.

References

1. Morrison, L. S. *et al.* Ecotin-like serine peptidase inhibitor ISP1 of *Leishmania major* plays a role in flagellar pocket dynamics and promastigote differentiation. *Cell Microbiol* **14**, 1271–1286 (2012).

REVIEWERS' COMMENTS

Reviewer #1 (Remarks to the Author):

The authors have done an excellent job of addressing concerns. Placing data in the context of the parasite life cycle highlights its significance.

Reviewer #2 (Remarks to the Author):

In this second revision of the manuscript, the authors have answered satisfactorily my comments regarding explaining in greater detail the proteomic analysis employed to screen for haptomonad-specific flagellar proteins. They have clarified how proteins were selected, primarily as those that exhibited no detectable spectrum counts in non-heptomonad parasites versus significant counts in attached heptomonads, and they validated the comparative proteomics dataset by demonstrating that flagellar proteins that were not expected to change significantly in abundance only differed by 0.5-2 fold between the two samples compared. These additional explanations are spelled out in the Materials and Methods section and address my previous comments.

Response to reviewers

Reviewer #1 (Remarks to the Author):

The authors have done an excellent job of addressing concerns. Placing data in the context of the parasite life cycle highlights its significance.

We are delighted by the reviewer's support and their appreciation of the additional data.

Reviewer #2 (Remarks to the Author):

In this second revision of the manuscript, the authors have answered satisfactorily my comments regarding explaining in greater detail the proteomic analysis employed to screen for haptomonad-specific flagellar proteins. They have clarified how proteins were selected, primarily as those that exhibited no detectable spectrum counts in non-heptomonad parasites versus significant counts in attached heptomonads, and they validated the comparative proteomics dataset by demonstrating that flagellar proteins that were not expected to change significantly in abundance only differed by 0.5-2 fold between the two samples compared. These additional explanations are spelled out in the Materials and Methods section and address my previous comments.

We would like to thank the reviewer for their appreciation of our additional explanations on the proteomic analysis that enhanced the manuscript.